# β-cell-specific deletion of *PFKFB3* restores cell fitness competition and physiological replication under diabetogenic stress

Jie Min[1,2], Feiyang Ma [ID] [3], Berfin Seyran[1], Matteo Pellegrini[3], Oppel Greeff[4], Salvador Moncada[5] & Slavica Tudzarova [ID] [1✉]

HIF1α and PFKFB3 play a critical role in the survival of damaged β-cells in type–2 diabetes while rendering β-cells non-responsive to glucose stimulation. To discriminate the role of PFKFB3 from HIF1α in vivo, we generated mice with conditional β-cell specific disruption of the *Pfkfb3* gene on a human islet pancreatic polypeptide (hIAPP$^{+/-}$) background and a high-fat diet (HFD) [PFKFB3$^{βKO}$ + diabetogenic stress (DS)]. PFKFB3 disruption in β-cells under DS led to selective purging of hIAPP-damaged β-cells and the disappearance of insulin- and glucagon positive bihormonal cells. PFKFB3 disruption induced a three-fold increase in β-cell replication as evidenced by minichromosome maintenance 2 protein (MCM2) expression. Unlike high-, lower DS or switch to restricted chow diet abolished HIF1α levels and reversed glucose intolerance of PFKFB3$^{βKO}$ DS mice. Our data suggest that replication and functional recovery of β-cells under DS depend on β-cell competitive and selective purification of HIF1α and PFKFB3-positive β-cells.

[1] Hillblom Islet Research Center, David Geffen School of Medicine, University of California, Los Angeles, CA, USA. [2] Department of Endocrinology, Union Hospital of Tongji Medical College Huazhong University of Science and Technology, Wuhan, Hubei, China. [3] Molecular Cell and Developmental Biology, College of Life Sciences, University of California Los Angeles, Los Angeles, CA, USA. [4] Department of Pharmacology, University of Pretoria, Pretoria, South Africa. [5] University of Manchester, Manchester, UK. ✉email: STudzarova@mednet.ucla.edu

Regenerative cell growth that exceeds physiological constraints is triggered by an increase in metabolic demands. It is a feature of adaptive processes, the outcome of which determines the development of various diseases such as diabetes[1,2]. Regenerative β-cell growth is triggered typically during pregnancy to compensate for the increased metabolic load of a developing foetus[3], or in non-diabetic, obese individuals in response to insulin resistance[4]. Diabetogenic stress (DS), which is caused by the accumulation of the toxic oligomers of islet-amyloid polypeptide (IAPP)[5–7], poses a particular challenge to long-lived and highly specialised pancreatic β-cells. With progressive accumulation of IAPP toxic oligomers, β-cells have to work harder to consolidate β-cell mass with a specialised function in order to maintain euglycaemia[2]. This balance typically falls short in type-2 diabetes (T2D)[8], however, it cannot be explained by a deficit in the β-cell mass, since β-cell mass is relatively preserved in T2D (between 35% and 76%)[6,9–12]. In contrast with β-cell mass, β-cell responsiveness to glucose declines even prior to the onset of T2D. This suggests that, although viable, most of the β-cells in T2D fail to adapt to the increasing metabolic demands under stress and are dysfunctional.

We have demonstrated previously that injured β-cells activate a highly conserved hypoxia-inducible factor-1 alpha (HIF1α) metabolic pathway and become entrapped via high rates of glycolysis that is disengaged from the TCA cycle, and which ensures the survival of injured β-cells while rendering them non-responsive to glucose[13]. The HIF1α effect on β-cell glucose metabolism was executed through 6-phosphofructo-2-kinase/fructose-2,6-biphosphatase 3 (PFKFB3), a key regulator of aerobic glycolysis under stress[13,14]. We reported that activation of HIF1α–PFKFB3 in rodent models of prediabetes preceded chronic glycolytic energy production, which further exacerbated the slip of β-cells into decompensation and diabetes progression[13,15,16].

Dysfunctional β-cells that can sustain damage with activation of HIF1α–PFKFB3 metabolic remodelling constitute one-fifth to one-third of all β-cells in humans with T2D[13]. They closely resemble suboptimal or unfit (compromised) cells that, by increasing aerobic glycolysis, have escaped the fitness quality control of the evolutionarily conserved cell competition[17–19]. Since the survival of injured β-cells in T2D depends on the glycolytic HIF1α–PFKFB3 pathway[13], targeting this pathway may differentially affect the fates of healthy and injured β-cells.

Based on the new evidence, we propose that the relationship between heterogeneous β-cell subpopulations that results from β-cells being subjected to injury is operative in β-cell replenishment by cellular replication. In our study, conditional knockout of PFKFB3 in adult β-cells under diabetogenic stress (human islet-amyloid polypeptide [hIAPP] and high-fat diet) (PFKFB3^βKO DS) led to selective elimination or purging of damaged β-cells and double-insulin- and glucagon-positive (bihormonal) cells and increased the replication of healthy β-cells. PFKFB3^βKO DS mice under lower diabetogenic stress exhibited improved glucose tolerance comparable with wild-type (WT) controls on HFD. However, in older PFKFB3^βKO DS mice under HFD for 13 weeks, glucose tolerance deteriorated, despite maintenance of the β-cell mass. Improved glucose tolerance at 8 weeks was correlated with diminished expression of HIF1α. Thus, β-cells with a HIF1α signature in the absence of PFKFB3 were probably responsible for the impairment of glucose tolerance at higher DS.

Human β-cells with a HIF1α signature (based on lactate-dehydrogenase, LDHA expression) from single-cell RNA sequencing (sc RNA Seq) analysis resembled double-insulin- and glucagon-positive cells in our mouse model of diabetes. The disruption of PFKFB3 induced the clearance of bihormonal cells at both higher and lower levels of DS with induction of comparable numbers of replicating β-cells. These results indicate a dichotomy in the control of β-cell replication and function by HIF1α and PFKFB3 under DS and offer a novel perspective that can be exploited in the therapy of diabetes.

## Results

### β-cells in PFKFB3^βKO mice under high diabetogenic stress demonstrate independent HIF1α expression and increased impaired glucose tolerance.

To study and dissect the role of PFKFB3 from HIF1α in the survival of damaged β-cells under diabetogenic stress in vivo, we generated mice with β-cell-specific conditional disruption of the *Pfkfb3* gene on a *hIAPP^+/−* background and exposed them to a high-fat diet (HFD) for 13 weeks (PFKFB3^βKO DS). Diabetogenic stress was deemed high since it involved insulin resistance (obesity) and exposure to misfolded proteins through hIAPP^+/− expression, that altogether with advanced age (44–50 weeks), are known as cumulative risk factors in diabetes[20–22].

PFKFB3^fl/fl hIAPP^+/− mice were born at the expected Mendelian ratio. From one week before the monitoring of mice up to the end of the experiment, there was no difference in the bodyweight among different experimental groups (Supplementary Fig. 1a–d). No difference was observed in the pancreas weight, but both spleen and liver showed lower weights in PFKFB3^βKO DS mice compared with the WT controls, although not reaching a significant difference (Supplementary Fig. 2a–c).

Efficient disruption of PFKFB3 expression was confirmed by PFKFB3 immunostaining of the pancreatic sections of the PFKFB3^βKO DS mice (Fig. 1a and Supplementary Information). Diabetogenic stress led to $33.9 \pm 6.4\%$ PFKFB3 immunolabelling of β-cells in PFKFB3^WT DS mice (**$p = 0.0015$), similar to the proportion of PFKFB3-positive β-cells previously reported in humans with T2D[22]. The proportion of PFKFB3-positive β-cells in WT mice was $3.7 \pm 1.9\%$, while in PFKFB3^βKO DS mice, it was successfully abolished and accounted for $1.0 \pm 0.8\%$ (***$p = 0.0006$, Fig. 1a, b).

Analysis of the metabolic performance of PFKFB3^βKO DS mice revealed glucose intolerance at both 9 (*$p = 0.0286$, $n = 4$) and 12 weeks after onset of the HFD (Fig. 2a, b and Supplementary Fig. 3a, b). Insulin-tolerance tests indicated higher insulin sensitivity in PFKFB3^βKO DS mice and, although insignificant, lower fasting-glucose levels, a difference that became diminished among the experimental groups at the later time points (Fig. 2c, d). C-peptide levels mirrored plasma-insulin levels and were lower in PFKFB3^βKO DS and PFKFB3^WT DS compared with the WT controls (*$p = 0.0283$ and *$p = 0.0464$, respectively) (Fig. 2e and g). No difference was observed in insulin clearance (Fig. 2h). Interestingly, although plasma-insulin levels were low in both PFKFB3^βKO DS and PFKFB3^WT DS mice, the latter had much higher plasma-glucagon levels. PFKFB3^βKO DS mice demonstrated a sharp reduction in glucagon levels in comparison with the PFKFB3^WT DS and the same levels as seen in the WT controls (Fig. 2f). These results, together with increased insulin sensitivity, suggested impaired insulin secretion in the PFKFB3^βKO DS mice.

To find out whether impaired glucose tolerance and insulin secretion in PFKFB3^βKO DS mice was linked to the presence of HIF1α, we immunostained pancreatic sections from all experimental groups with HIF1α antibody. Expression of HIF1α increased to $18.4 \pm 4.2\%$ in β-cells from PFKFB3^WT DS mice compared with WT controls (*$p = 0.034$). In the PFKFB3^βKO DS mice, $14.2 \pm 3.8\%$ of all β-cells (Fig. 1c, d) showed HIF1α immunopositive cytoplasm and nucleus (Supplementary Information) in the absence of PFKFB3. This clearly indicates that PFKFB3 knockout triggered independent HIF1α expression in response to high diabetogenic stress.

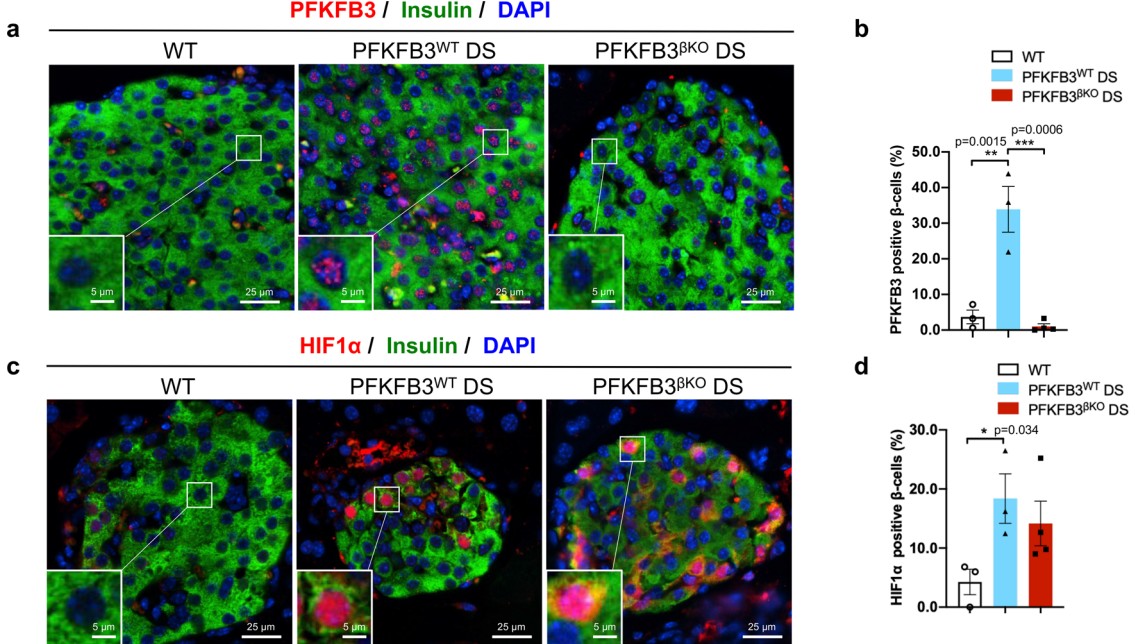

**Fig. 1 HIF1α is upregulated in PFKFB3$^{βKO}$ DS mice under high diabetogenic stress. a** Representative immunofluorescence images of islets from WT, PFKFB3$^{WT}$ DS, and PFKFB3$^{βKO}$ DS mice at 13 weeks HFD immunostained for PFKFB3 (red), insulin (green) and nuclei (blue). **b** Quantification of images in (**a**) (**\*\****p* = 0.0015, \*\*\**p* = 0.0006). **c** Representative immunofluorescence images of islets from WT, PFKFB3$^{WT}$ DS and PFKFB3$^{βKO}$ DS mice at 13 weeks HFD immunostained for HIF1α (red), insulin (green) and nuclei (blue). **d** Quantification of images in (**c**) (*n* = 3, *n* = 4 for PFKFB3$^{βKO}$ DS–independent animals, SEM \**p* = 0.034).

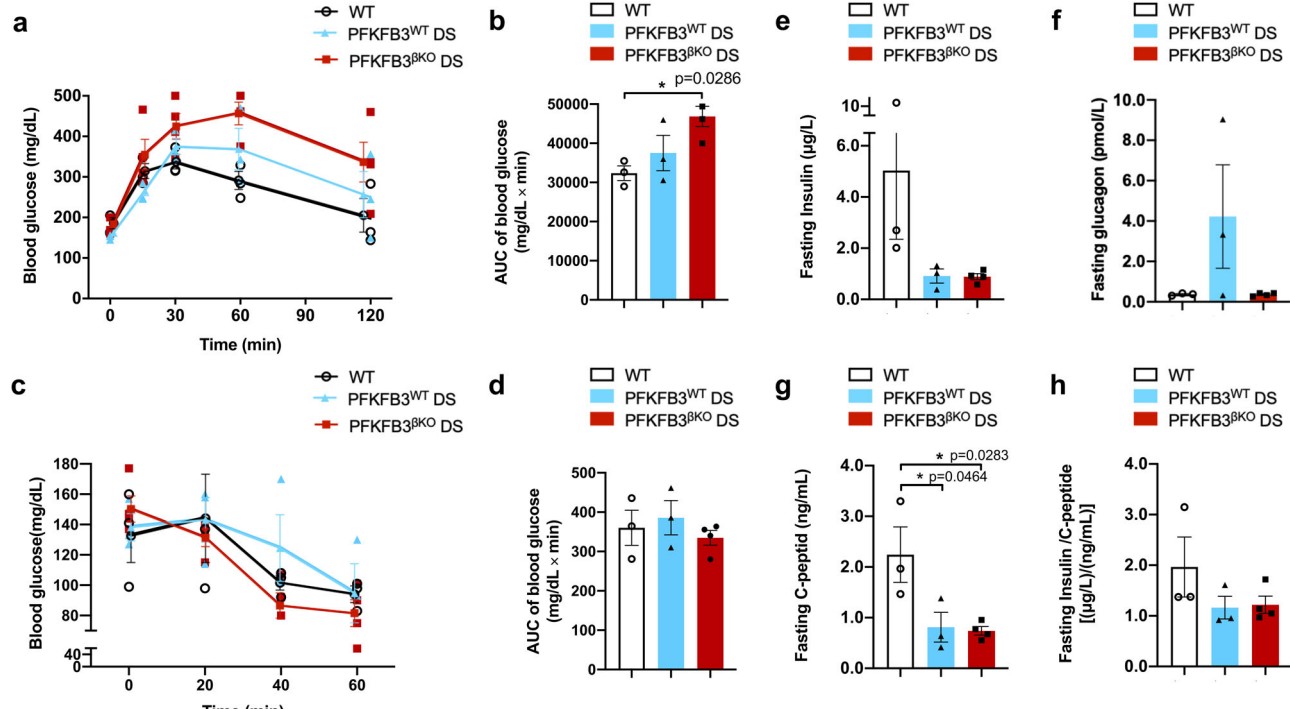

**Fig. 2 PFKFB3$^{βKO}$ DS high mice demonstrate increased impairment of glucose tolerance and similar insulin- but reduced glucagon-plasma levels relative to PFKFB3$^{WT}$ DS mice. a** Intraperitoneal glucose-tolerance test (IP-GTT) at nine weeks post onset of high-fat diet (HFD). **b** Quantification of the area under the curve (AUC) as mg/dl x min in the experimental groups shown in (**a**) (\**p* = 0.0286). **c** Insulin-tolerance test (ITT) at 10 weeks after onset of HFD. **d** Quantification of the AUC as mg/dl x min in the experimental groups shown in (**c**). **e** Fasting-plasma insulin. **f** Fasting-plasma glucagon. **g** Fasting C-peptide and (**h**) fasting-insulin/C-peptide ratio (12 weeks post onset of HFD) (*n* = 3, *n* = 4 for PFKFB3$^{βKO}$ DS-independent animals, SEM).

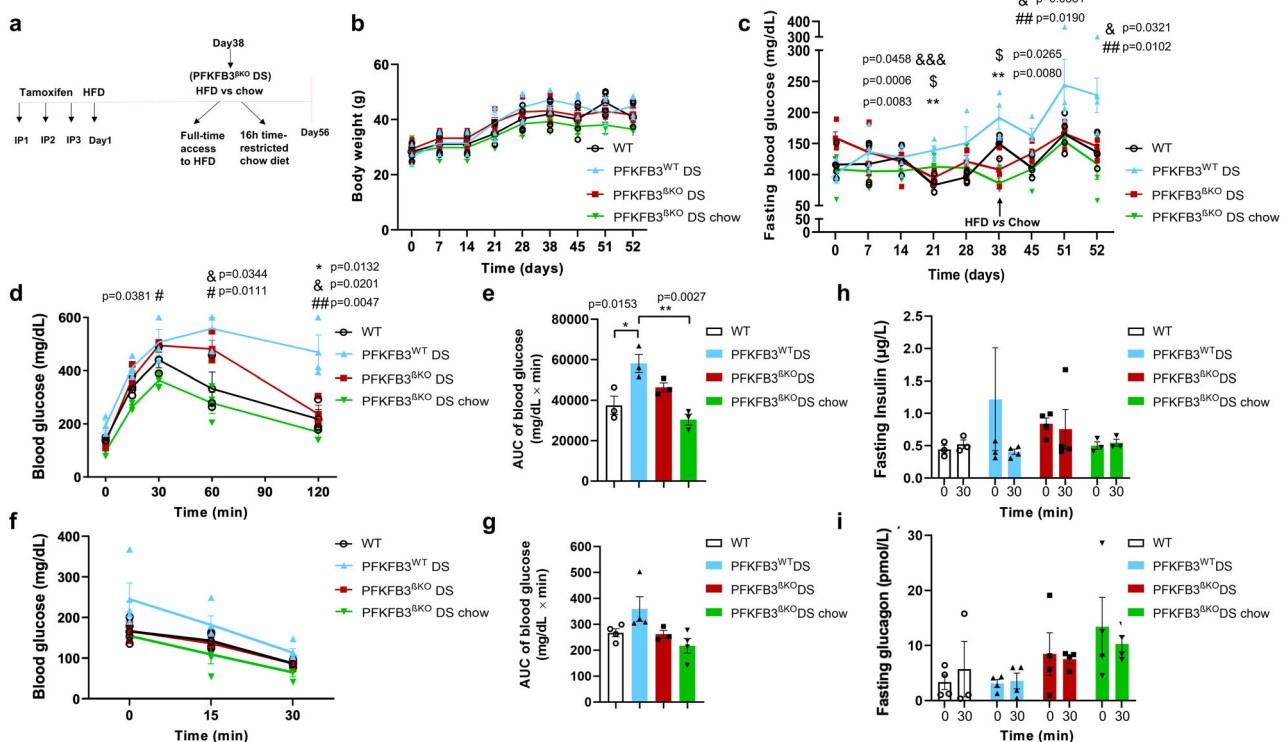

**Fig. 3 PFKFB3βKO DS low mice demonstrate improved glucose tolerance and insulin sensitivity relative to PFKFB3WT DS low mice. a** Scheme of experimental paradigm with lower diabetogenic stress. Intraperitoneal injections 1, 2 and 3 (IP1, 2, 3). Days after onset of HFD are indicated. **b** Body weight of mice in the experimental groups. **c** Weekly fasting blood glucose in the indicated experimental groups (*p values for significant difference between PFKFB3βKO DS and PFKFB3WT DS, #p values for significant difference between PFKFB3βKO DS chow and PFKFB3WT DS mice and $p values for significant difference between PFKFB3WT DS and WT are indicated) (see individual values in Supplementary Data File 6). **d** Intraperitoneal glucose tolerance test (IP-GTT) at 8 weeks post onset of high-fat diet (HFD) (n = 3 independent experiments, each experiment n = 4 independent animals, SEM) (see individual values in Supplementary Data File 6). **e** Quantification of the area under the curve (AUC) as mg/dL x min in the experimental groups in (**d**) (*p = 0.0153, **p = 0.0027, n = 4 independent animals, SEM). **f** Insulin-tolerance test at 8 weeks after onset of HFD in all groups except PFKFB3βKO DS chow (4 weeks of HFD and 4 weeks of 16 h restricted feeding with chow diet). **g** Quantification of the area under the curve (AUC) as mg/dL x min in the experimental groups in (**f**). **h** Fasting plasma insulin and (**i**) fasting plasma glucagon (8 weeks post onset of HFD) (n = 3 for PFKFB3βKO DS chow, n = 4 for PFKFB3βKO DS–independent animals, SEM).

Given that HIF1α can metabolically reprogramme β-cells disrupting the glucose-secretion coupling, we next sought to discover the origin and role of HIF1α expression in the impaired metabolic performance of PFKFB3βKO DS mice.

**PFKFB3βKO DS mice under lower diabetogenic stress show improved glucose tolerance linked to reduced expression of HIF1α.** To address the origin of the independent HIF1α expression that occurred in the absence of PFKFB3, we repeated the experiment in younger mice (7–10 weeks of age) and under 8-week exposure to HFD (lower diabetogenic stress). Four weeks after tamoxifen injection and exposure to HFD, the PFKFB3βKO DS mice were split into two groups: one group (n = 4) continued HFD, and another group was switched to chow diet with access to food 16 hours per day three times a week for another 4 weeks (Fig. 3a). Over the course of time, the body weight remained unaltered between the groups (Fig. 3b).

While showing progressive elevation with the time of the exposure to diabetogenic stress in PFKFB3WT DS mice, the fasting blood glucose remained lower in the PFKFB3βKO DS and WT controls compared with the PFKFB3WT DS mice (**p = 0.0083 and &&&p = 0.0006, at day 21 and **p = 0.0080 at day 38, respectively) (Fig. 3c). Fasting glucose levels improved in the PFKFB3βKO DS mice that were switched to time-restricted chow diet and remained lower similar to the trend in the WT controls

relative to the PFKFB3WT DS mice (#p = 0.0190 and &p = 0.0381, respectively, at day 45 and #p = 0.0102 and &p = 0.0321 at day 52) (Fig. 3c).

After being injected with glucose bolus, WT mice exhibited a drop in blood glucose at 60 and 120 min (&p = 0.0344 and &p = 0.0132, respectively); PFKFB3βKO DS chow mice exhibited improved glucose tolerance from 30 to 120 min (#p = 0.0381, #p = 0.0111 and ##p = 0.0047, respectively), and PFKFB3βKO DS exhibited improved glucose tolerance at 120 min (*p = 0.0132) relative to PFKFB3WT DS mice (Fig. 3d, e). ITT revealed similar insulin sensitivity between the PFKFB3βKO DS and the WT controls, while increased insulin sensitivity in both groups compared with the PFKFB3WT DS mice. As expected, the PFKFB3βKO DS mice on a chow diet had the highest insulin sensitivity (Fig. 3f, g). Unlike in all other groups, only in PFKFB3WT DS mice, plasma-insulin levels dropped at 30 minutes after glucose stimulation (Fig. 3h). In the PFKFB3βKO DS group, the basal insulin was higher compared with the WT and the PFKFB3βKO DS under chow diet. Plasma-glucagon levels were lower in the PFKFB3WT DS mice compared with the other groups (Fig. 3i). HIF1α was upregulated in PFKFB3WT DS mice to a similar extent as observed at 13 weeks of HFD (**p = 0.0031). To our surprise metabolic improvement in both PFKFB3βKO DS and PFKFB3βKO DS chow mice correlated with significantly reduced levels of HIF1α as measured in nucleus and cytoplasmic compartments together, and cytoplasmic compartment only (both

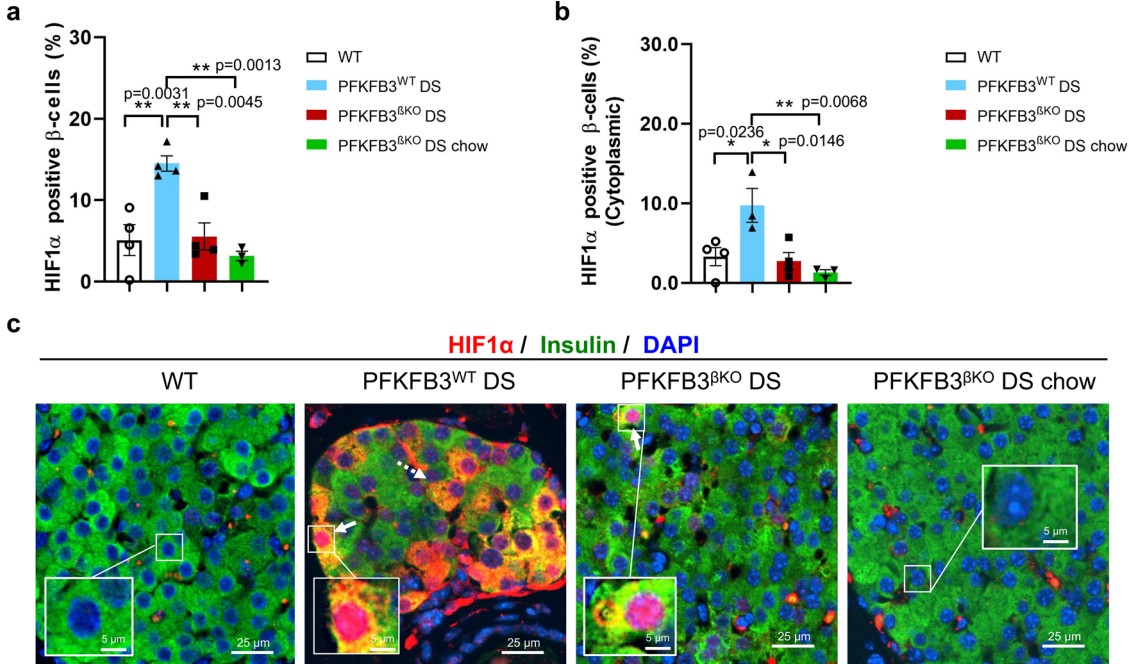

**Fig. 4 HIF1α is downregulated in PFKFB3^βKO DS mice under lower diabetogenic stress. a** Quantification of HIF1α (total nuclear and cytoplasmic) positive β-cells in the indicated experimental groups (PFKFB3^WT DS relative to WT; PFKFB3^WT DS relative to PFKFB3^βKO DS and relative to PFKFB3^βKO DS chow, n = 4, n = 3 for PFKFB3^βKO DS chow-independent animals, SEM, **p = 0.0031, **p = 0.0045 and **p = 0.0013, respectively). **b** Quantification of HIF1α-(cytoplasmic only) positive β-cells in the indicated experimental groups (PFKFB3^WT DS relative to WT; PFKFB3^WT DS relative to PFKFB3^βKO DS and relative to PFKFB3^βKO DS chow, n = 4, n = 3 for PFKFB3^βKO DS chow-independent animals, SEM, **p = 0.0236, **p = 0.0146 and **p = 0.0068, respectively). **c** Representative immunofluorescence images of islets from WT and PFKFB3^βKO DS mice at 8 weeks of HFD with or without switch to chow diet (PFKFB3^βKO DS chow) immunostained for HIF1α (red), insulin (green) and nuclei (blue).

nuclear and cytoplasmic 2.1 ± 0.8% and 1.8 ± 0.4%; cytoplasmic only 2.7 ± 1.1% and 1.3 ± 0.3%, respectively) (**p = 0.0045 and **p = 0.0013, and cytoplasmic *p = 0.0146 and **p = 0.0068, respectively) (Fig. 4a–c). These results clearly indicated that the decompensation of the metabolic recovery in PFKFB3^βKO DS mice depended on the magnitude of the diabetogenic stress reflected in the HIF1α levels. High diabetogenic stress resulted in metabolic decompensation in the PFKFB3^βKO DS mice and was linked to high levels of HIF1α independent of PFKFB3. Younger age and shorter exposure to HFD resulted in a metabolic compensation after the PFKFB3 disruption, with glucose tolerance comparable to the WT controls and diminished HIF1α expression levels.

**β-cells with HIF1α signature in humans with T2D show a bihormonal status, express disallowed genes and genes that confer immaturity.** HIF1α immunostaining in the pancreatic sections of old PFKFB3^βKO DS mice under 13 weeks of HFD affected about 14% of all β-cells. We posited that the lingering expression of HIF1α, independent from PFKFB3, is instrumental in the metabolic dysfunction of PFKFB3^βKO DS mice under high diabetogenic stress (Fig. 1c, d). As such, reduction of the diabetogenic stress or switch to chow diet improved metabolic function coinciding with the reduced expression levels of HIF1α.

To elucidate how HIF1α-positive β-cells may contribute to the metabolic decompensation, we next investigated the genetic make-up of the HIF1α-positive β-cells in human T2D. We used single-cell (sc) RNA-Seq data from humans with T2D and nondiabetics available in a public repository[23]. First, we analysed the quality and validated the sc RNA-Seq data (Supplementary Figs. 4 and 5). Pancreatic cells from healthy and T2D donors were reclustered (umap_cluster) and annotated to the specific cell types based on the gene markers such as insulin (INS) for β-cells (umap_celltype, Fig. 5a±d).

Nine different cell types were identified and the differentially expressed genes are presented in Fig. 5c and Supplementary Figs. 6–8. β-cells were then separated into those from healthy and T2D conditions (umap_disease, Fig. 5a). Using INS as a β-cell marker, we identified two clusters of β-cells, cluster 1 and cluster 7, while clusters 2–6, 8 and 9 referred to other pancreatic cell types (Fig. 5b, c). Composition in clusters 6 (α-cell subpopulation), 7 (β-cell subpopulation) and 8 (δ-cells) differed the most between healthy and T2D donors (Supplementary Fig. 6). Comprehensive tables listing all differentially expressed genes between the groups are presented in Supplementary Data Files 1–4 and Supplementary Tables 1–3. Since HIF1α is mainly regulated in a posttranslational way, we distinguished β-cell subpopulations based on the expression or not of the lactate-dehydrogenase A (LDHA), a HIF1α transcriptional target from aerobic glycolysis (Fig. 5d). For each condition, and based on LDHA expression, cells were split into LDHA-positive and LDHA-negative cells, and differential expression analysis was performed between the two groups (healthy and T2D) (Fig. 6a–d)[24,25]. To our surprise, LDHA-positive β-cells almost completely overlapped with cluster-7 β-cells (Fig. 5c, d), and, independent of the disease state, co-aggregated with genes relevant to metabolism (Supplementary Tables 4 and 5), such as LDHA, glucokinase (GK), phosphofructokinase-1 platelet type (PFKFP), pyruvate dehydrogenase kinase 4 (PDK4), or genes relevant for insulin secretion such as FBP1 via phosphoenolpyruvate pool, or identity such as glucagon (GCG, pro-α-cell identity) and Aristaless related homeobox (ARX, pro-α-cell identity) and INS (lower expression, β-cell identity). PFKFB3 levels were low to undetectable and therefore they could not be correlated with LDHA-positive versus -negative β-cells (Supplementary Fig. 9).

LDHA-positive or cluster-7 β-cells showed an immature phenotype in line with upregulation of genes such as aldehyde dehydrogenase 1A1 (ALDH1A1)[26]. Interestingly, although these

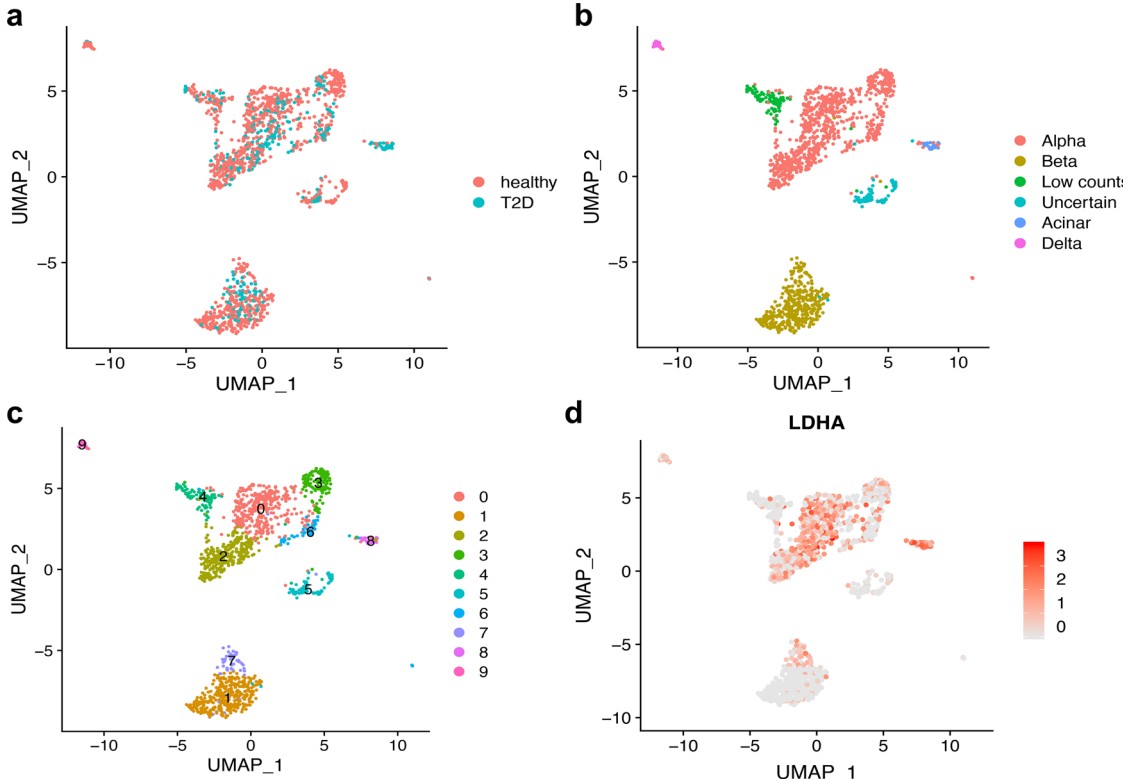

**Fig. 5 Cluster-7 β-cell subpopulation overlaps with LDHA-positive β-cells with HIF1α signature. a** UMAP plot of cluster distribution of 1,482 pancreatic cells coloured by disease conditions (health and T2D). **b** UMAP plot of cluster distribution and coloured by cell-type annotations [α-cells (alpha), β-cells (beta), cells with low counts, cells with non-verified identity, acinar cells and ductal cell]. **c** UMAP plot coloured by unsupervised clusters. **d** UMAP plot showing LDHA expression. The colour scale represents normalised expression level of the gene. Published sc RNA-Seq in[23] was presented in a quantitative way to compare nine identified pancreatic-cell subpopulations in the top differentially expressed genes.

cells expressed more pro-apoptotic BID, they expressed at the same time the inhibitor of apoptosis Spark/Osteonectin (SPOCK3), a key to anti-apoptotic defence of cell subpopulations with lower fitness and thus survival features according to cell-fitness competition (Fig. 6b–d)[17].

For comparison, *LDHA*-positive versus *LDHA*-negative α-cells showed top enrichment in gene pathways related to aerobic glycolysis and metabolism (Supplementary Data File 5a, b and Supplementary Table 6).

These results pinpointed that in humans with T2D, a fraction of β-cells (*LDHA*-positive cells overlapping with cluster 7 β-cells) possess a genetic signature with reduced INS and increased GCG and ARX expression. Insight into gene overrepresentation by Ingenuity Pathway Analysis (IPA) revealed that the difference between the significantly altered genes in clusters 1 and 7, and between *LDHA*-positive and -negative cells, can be recapitulated by liver-X/retinoid-X receptor (LXR/RXR)[27,28], indicating this upstream regulator as a part of the epistatic HIF1α non-canonical metabolic pathway (Supplementary Table 7). Moreover, the String analysis clearly indicated that while differences between clusters 1 and 7, as well as *LDHA*-negative and -positive β-cells, were well preserved in health, these differences were strongly reduced in T2D. These data suggested that in T2D, clusters of β-cells begin to resemble each other (Supplementary Figs. 10a, b and 11a, b, and Venn diagrams in Fig. 6e, f). When compared between T2D and health, differentially expressed genes in either cluster 1 or *LDHA*-negative β-cells showed major overlap. While a difference was observed in cluster 1 (Supplementary Fig. 12a, b), no difference was observed in cluster 7 or *LDHA*-negative β-cells when health was compared with T2D. Collectively, given that in T2D, clusters 7 and 1 become more resembling, and cluster 7

does not change, it implies that cluster 1 undergoes changes to become similar to cluster 7 with HIF1α signature.

**Insulin- and glucagon double-positive (bihormonal) cells in diabetic mice resemble human β-cells with HIF1α signature.** First, we established that human β-cells with HIF1α signature are dysfunctional due to activation of disallowed genes such as *LDHA*, immature (*ALDH1A1*) and bihormonal status (INS- and GCG positive). To find a complementary β-cell population to cluster 7 or *LDHA*-positive cells relative to their bihormonal status, we double-stained pancreatic sections from our experimental mice with specific insulin and glucagon antibodies.

Diabetogenic stress at 8 weeks increased sevenfold and at 13 weeks increased twice the number of double-positive (bihormonal) cells in PFKFB3^WT DS compared with WT controls (3.8 ± 0.7% relative to 0.5 ± 0.3% and 5.7 ± 2.8% relative to 2.7 ± 1.8%, respectively) (Fig. 7a,d and Supplementary Fig. 13). This ratio reflected the increased proportion of bihormonal cells over time in WT mice under HFD (from 0.5 ± 0.3% to 2.7 ± 1.8%, respectively). At 8 and 13 weeks of HFD, the proportion of bihormonal cells in PFKFB3^βKO DS mice was consistently low (1.3 ± 0.9% and 0.8 ± 0.3%, respectively) indicating that the attrition of bihormonal cells was not gradual over time after PFKFB3 knockout (Fig. 7a, d). This was emphasised with the finding of bihormonal cells not overlapping with cleaved caspase 3 at any time point or experimental group, including the PFKFB3^βKO DS mice, indicating that PFKFB3 is involved either in the generation or the sustenance but not in the control of the bihormonal cell death (Fig. 7g). The proportion of bihormonal cells in PFKFB3^WT DS over time closely matched the increase in β-cells in PFKFB3^βKO DS mice under high-fat- or chow diet

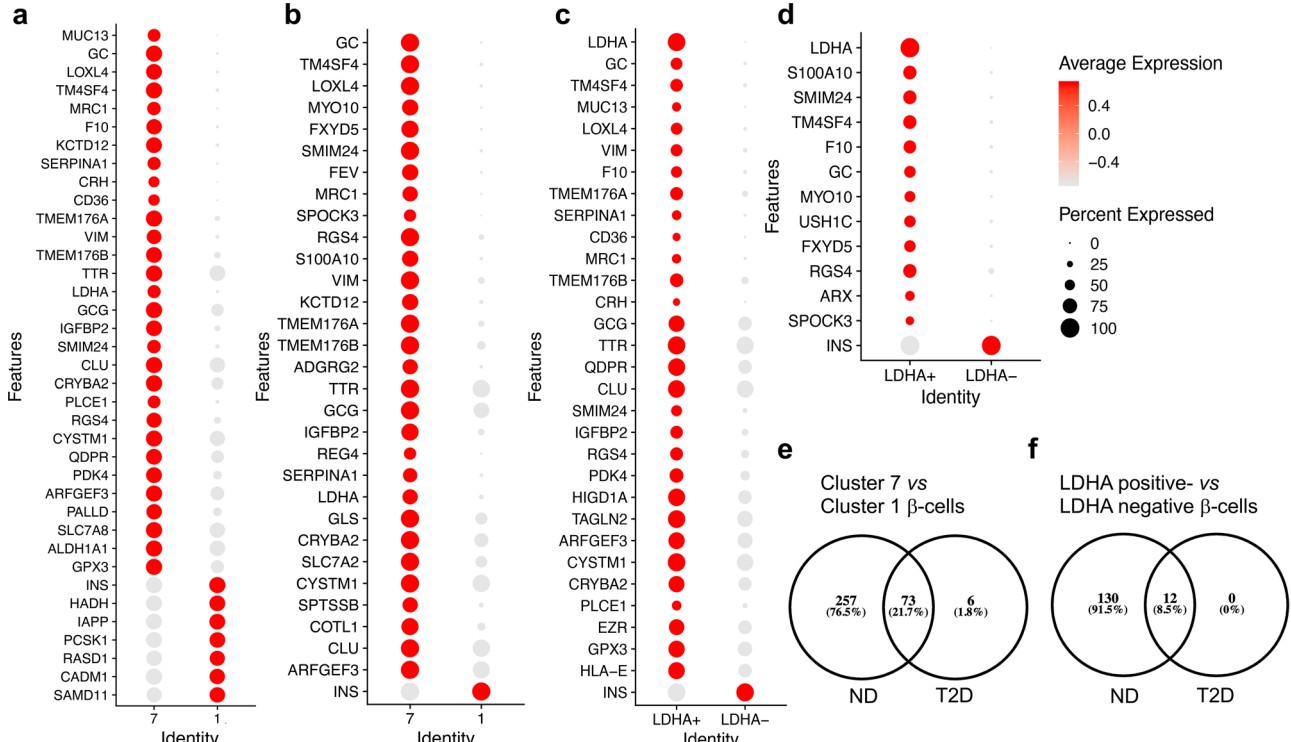

**Fig. 6 Differential-expression analysis comparing LDHA-positive and LDHA-negative β-cells from healthy and T2D donors. a** Dot plot showing the differentially expressed genes between cluster 7 and cluster 1 β-cells in non-diabetic donors (ND). The size of the dots represents the percentage of cells the gene was detected in. The colour scale represents the scaled expression of the gene in the two groups. The Wilcoxon rank-sum test was performed in the differential expression analysis, and the Benjamini–Hochberg procedure was applied to adjust the false-discovery rate. Genes with adjusted $p$-value less than 0.05 were considered significantly differentially expressed. **b** Dot plot showing the differentially expressed genes between cluster 7 and cluster 1 β-cells in T2D. **c** Dot-plot showing the differentially expressed genes between LDHA-positive and LDHA-negative β-cells from non-diabetic donors (ND). **d** Dot plot showing the differentially expressed genes between LDHA-positive and LDHA-negative β-cells from T2D. **e** Venn diagram from the results presented under (**a**) and (**b**). (**f**) Venn diagram from the results presented under (**c**) and (**d**).

($*p = 0.0282$ and $*p = 0.0320$, respectively) (Fig. 7c, f). This indicated that cells with double insulin and glucagon identity either shifted to replicating β-cells or their culling contributed to increased β- over α-cell ratio in PFKFB3βKO DS mice. Reversal in favour of β- relative to α- cells was observed at 8 week and 13 week exposure to HFD, reaching about 95% in PFKFB3βKO DS mice (Fig. 7c, f). The opposite was true for α-cells in PFKFB3βKO DS mice at both 8 and 13 weeks of HFD relative to all α-, β- and bihormonal cells together (Fig. 7b, e). This ratio reached control levels ($3.7 \pm 0.9\%$ and $3.7 \pm 0.7\%$ at 8 weeks and $4.7 \pm 0.2\%$ and $4.1 \pm 1.5\%$ at 13 weeks) and was reduced in PFKFB3βKO DS compared with PFKFB3WT DS mice ($3.7 \pm 0.9\%$ relative to $9.0 \pm 2.1\%$ and $4.7 \pm 0.2$ relative to $6.2 \pm 0.5\%$, respectively). Of note, the use of a HFD versus a chow diet led to a decrease in α-cell count over time in PFKFB3WT DS mice (from $9.0 \pm 2.1\%$ to $6.2 \pm 0.5\%$, Fig. 7b, e) in line with α-cell hypotrophy reported in obesity[29]. In addition, the bihormonal cells were present in our experimental mice exposed to the HFD and were not detected in WTchow controls or hIAPP+/+ (hTG) on a chow diet used as negative controls, neither in prediabetic nor in diabetic mice (Supplementary Fig. 14a, b).

Next, we investigated how the dynamics between bihormonal-, α- and β-cells was reflected in the β-cell mass and sought to determine whether β-cell mass is implicated in the β-cell decompensation in PFKFB3βKO DS mice under high diabetogenic stress as suggested from our previous in vitro studies[13].

**β-cells in PFKFB3βKO DS mice demonstrate increased cell turnover and replication.** The β-cell fractional area and mass

were unaltered among the experimental groups (Fig. 8a, b). To investigate the growth dynamics that led ultimately to comparable β-cell mass between the PFKFB3βKO DS and PFKFB3WT DS mice, we performed TUNEL staining to measure past cell death (Fig. 8c) and cleaved caspase-3 immunostaining to measure active β-cell death[30] (Fig. 8e, f). According to the TUNEL analysis, past β-cell death was increased in the PFKFB3βKO DS mice relative to the WT mice (Fig. 8c) and no difference was measured in the ongoing cell death by cleaved caspase 3 (Fig. 8f). To our surprise, the β-/α-cell ratio was increased in the PFKFB3βKO DS relative to the PFKFB3WT DS mice (Fig. 8d) and this posed a question regarding its origin.

To further elucidate whether the increase in the β-/α-cell ratio relied on the increased generation of β-cells, we performed immunolabelling with the early replication initiation marker, minichromosome-maintenance protein 2 (MCM2)[31,32]. Our data showed that β-cells from the PFKFB3βKO DS mice exhibited a three-fold increase in MCM2 labelling ($5.3 \pm 0.8\%$, $*p = 0.029$), indicating increased β-cell replication compared with the PFKFB3WT DS mice ($1.9 \pm 0.04\%$) and similar to the WT controls ($7.0 \pm 1.3\%$, Fig. 9a, b). This was present also at 8 weeks of HFD (Fig. 9c). Despite the increase in the rate of both cell death and β-cell replication in the PFKFB3βKO DS mice, the β-cell fractional area was comparable in all three groups (Fig. 8a). Thus, the increment in β-cell replication in the PFKFB3βKO DS mice appeared to maintain β-cell mass, despite the increased cell death and initial loss of damaged β-cells (measured by TUNEL assay) in the absence of PFKFB3.

To clarify whether replicating β-cells possessed any residual injury, we made use of the specific marker of hIAPP-incurred

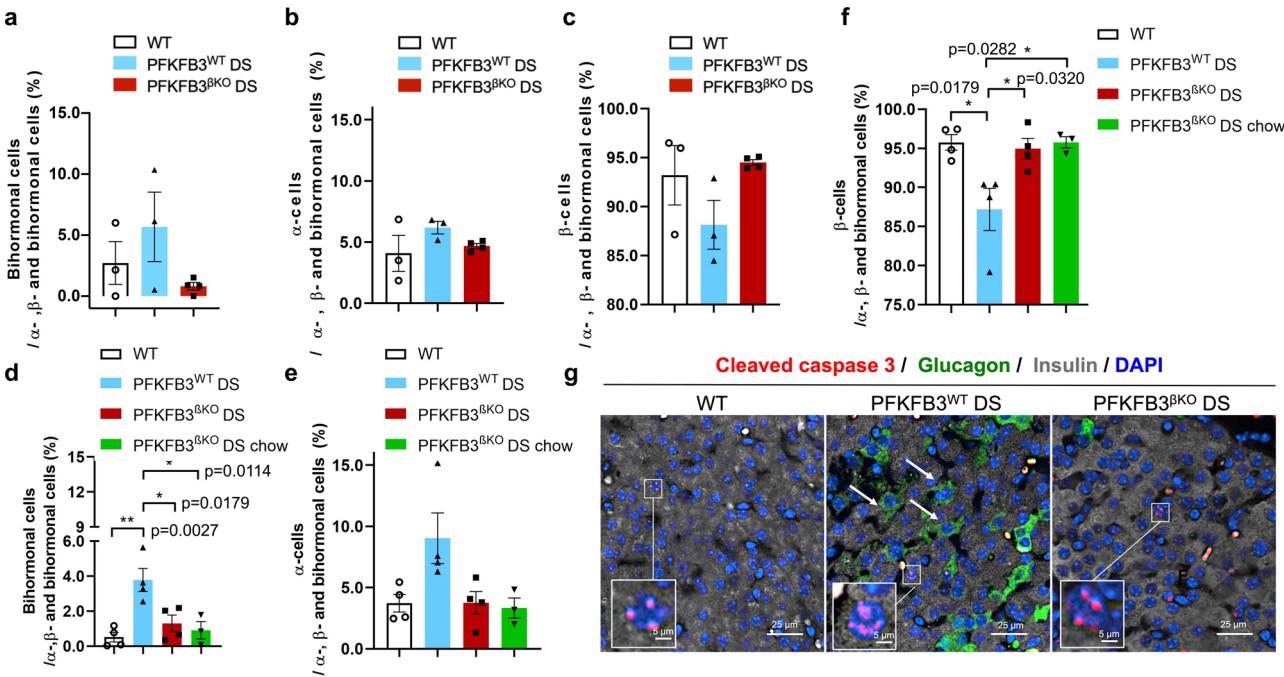

**Fig. 7 PFKFB3^βKO DS mice show a decrease of double- insulin and glucagon positive cells at lower and higher diabetogenic stress. a** Quantification of the ratio between bihormonal (insulin- and glucagon positive) cells relative to all α-, β- and bihormonal cells (%) at 13 weeks of HFD. **b** Quantification of the ratio between α-cells relative to all α-, β- and bihormonal cells (%) at 13 weeks of HFD. **c** Quantification of the ratio between β-cells relative to all α-, β- and bihormonal cells (%) at 13 weeks of HFD. **d** Quantification of the ratio between bihormonal (insulin- and glucagon positive) cells relative to all α-, β- and bihormonal cells (%) at 8 weeks HFD (**$p = 0.0027$, *$p = 0.0179$, *$p = 0.0114$, $n = 4$ independent animals, SEM). **e** Quantification of the ratio between α-cells relative to all α-, β- and bihormonal cells (%) at 8 weeks of HFD. **f** Quantification of the ratio between β-cells relative to all α-, β- and bihormonal cells (%) at 8 weeks of HFD (*$p = 0.0179$, *$p = 0.0320$, *$p = 0.0282$, $n = 4$ independent animals, SEM). **g** Representative immunofluorescence images of islets from WT, PFKFB3^WT DS and PFKFB3^βKO DS mice at 13 weeks of HFD co-immunostained for cleaved caspase-3 (red), insulin (grey), glucagon (green) and nuclei (blue).

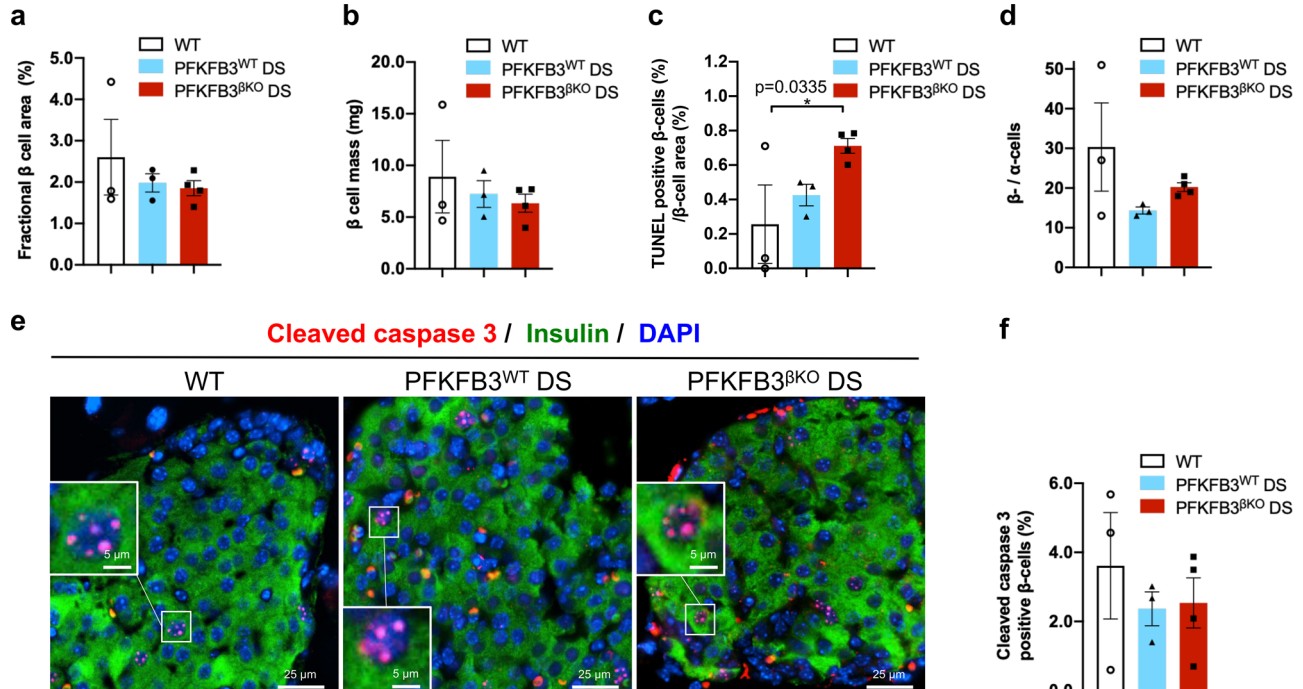

**Fig. 8 PFKFB3^βKO DS high mice show increased β-/α-cell ratio in spite of the increase in the β-cell death relative to PFKFB3^WT DS mice. a** Quantification of fractional β-cell area (%). **b** Quantification of β-cell mass (mg). **c** Quantification of β-cell death as measured by labelling with TUNEL assay (%) represented relative to fractional β-cell area (*$p = 0.0335$). **d** Quantification of β-cell relative to α-cell number in indicated experimental groups. **e** Representative immunofluorescence images of islets from WT, PFKFB3^WT DS and PFKFB3^βKO DS mice immunostained for cleaved caspase-3 (red), insulin (green) and nuclei (blue). **f** Quantification of images under (**e**) ($n = 3$, $n = 4$ for PFKFB3^βKO DS-independent animals, SEM).

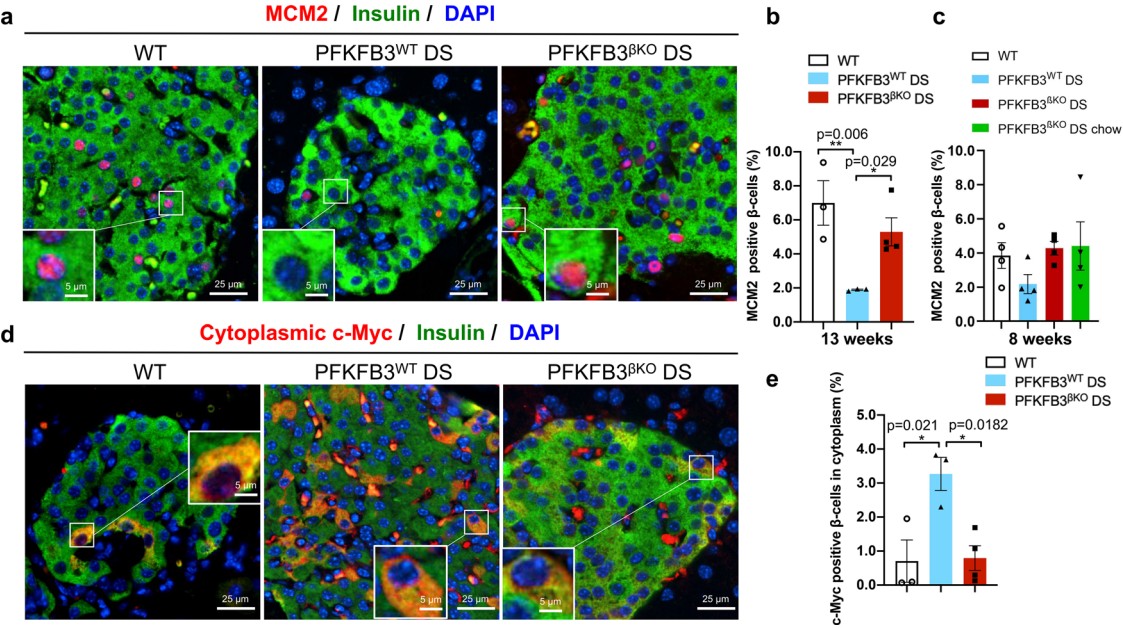

**Fig. 9 PFKFB3$^{\beta KO}$ DS mice show increased rates of healthy β-cell replication compared with PFKFB3$^{WT}$ DS mice. a** Representative immunofluorescence images of islets from WT, PFKFB3$^{WT}$ DS and PFKFB3$^{\beta KO}$ DS at 13 weeks of HFD immunostained for MCM2 (red), insulin (green) and nuclei (blue). **b** Quantification of images shown under (**a**) (**$p = 0.006$, *$p = 0.029$). **c** Quantification of the total of all medium and strong expressing MCM2 positive β-cells in the islets from WT, PFKFB3$^{WT}$ DS and PFKFB3$^{\beta KO}$ DS at 8 weeks of HFD and PFKFB3$^{\beta KO}$ DS chow (4 weeks under 16 h restricted chow diet following 4 weeks of HFD). **d** Representative immunofluorescence images of islets from PFKFB3$^{WT}$ DS and PFKFB3$^{\beta KO}$ DS at 13 weeks immunostained for c-Myc (red), insulin (green) and nuclei (blue). **e** Quantification of cytoplasmic c-Myc indicates cells undergoing hIAPP-induced calpain activation (damage) as revealed by immunostaining in (**d**) (*$p = 0.021$; *$p = 0.0182$, $n = 3$, $n = 4$ for PFKFB3$^{\beta KO}$ DS-independent animals, SEM).

damage in β-cells—the cytoplasmic accumulation of the calpain-mediated truncation of c-Myc[33]. No cytoplasmic c-Myc was detected at 8 weeks of HFD in any of the experimental groups (Supplementary Fig. 15a). However, at 13 weeks, our analysis revealed an increase of cytoplasmic c-Myc truncation in the PFKFB3$^{WT}$ DS mice ($3.3 \pm 0.5\%$, *$p = 0.021$), but reversal to the WT control levels in the PFKFB3$^{\beta KO}$ DS mice ($0.7 \pm 0.6\%$ and $0.8 \pm 0.4\%$, respectively, Fig. 9d, e). Interestingly, C/EBP homologous-protein (CHOP) immunostaining indicated an ongoing endoplasmic reticulum (ER) stress in PFKFB3$^{WT}$ DS mice at 8 weeks, that was low to undetectable in WT controls or PFKFB3$^{\beta KO}$ DS mice (Supplementary Fig. 15b, c). These data indicated that lower versus higher diabetogenic stress can be distinguished by ongoing ER damage in the absence of misfolded protein stress. Moreover, at 13 weeks, the truncated c-Myc did not show any overlap (below 0.6%) with MCM-2, indicating that MCM-2 positive replicating β-cells were free of damage (Supplementary Fig. 16a–d). The matching proportion of bihormonal cells in PFKFB3$^{WT}$ DS and replicating cells in PFKFB3$^{\beta KO}$ DS mice at both time points (Figs. 7a,d and 9b,c) indicated that healthy β-cells contributed to the increment in replication, in relation to increased clearance of bihormonal- and hIAPP-damaged β-cells.

## Discussion

In this study, we demonstrate that the specific β-cell disruption of the *Pfkfb3* gene in adult mice under diabetogenic stress leads to restoration of islet mass and function with an extent dependent on the level of the stress and the expression of HIF1α.

Under high diabetogenic stress (52 weeks of age, 13 weeks of HFD), in PFKFB3$^{\beta KO}$ DS mice, the maintenance of the β-cell mass is not followed by the restoration of β-cell function. Both β-cell mass and function can be compensated during lower diabetogenic stress (18 weeks of age and 8 weeks of HFD). While in

both cases PFKFB3 was specifically depleted from β-cells, what appeared to be correlating with the decompensation was the independent expression of HIF1α. HIF1α was comparably expressed in the PFKFB3$^{WT}$ DS- and in the PFKFB3$^{\beta KO}$ DS mice under high DS, but it was low in the young PFKFB3$^{\beta KO}$ DS mice under 8 weeks of HFD or chow diet, the groups where glucose tolerance and insulin sensitivity were improved. This clearly delineates the separate role HIF1α plays in controlling β-cell function compared with PFKFB3.

In our experimental paradigm, misfolded protein stress followed the ER stress after prolonged exposure to HFD (13 weeks) in older mice. We monitored hIAPP-injured β-cells by measuring the extent of the calpain-mediated truncation of the cytoplasmic c-Myc[33]. Calpain was previously reported to directly reflect hIAPP misfolded protein toxicity[34,35]. Unlike ER-stress marker CHOP that was increased early, cytoplasmic c-Myc was not detected before 13 weeks of HFD, and increased cytoplasmic c-Myc was reversed in PFKFB3$^{\beta KO}$ DS mice to the barely detectable levels similar to WT controls. These results suggested that at 13 weeks of HFD, the sustained survival of a proportion of stressed β-cells was due to PFKFB3-independent HIF1α expression probably triggered by the misfolded protein stress.

Interestingly, the number of bihormonal cells in PFKFB3$^{WT}$ DS mice did not vary between the experimental groups at the two time points (8 weeks and 13 weeks of HFD). Moreover, the number of bihormonal cells in the PFKFB3$^{WT}$ DS mice closely matched the number of replicating β-cells in the PFKFB3$^{\beta KO}$ DS mice. Bihormonal cells did not overlap with cleaved caspase-3-labelled cells in any experimental group and at any experimental time. These results indicated that rather than for the survival, PFKFB3 expression was probably necessary for the generation of bihormonal cells. Thus, our data suggest that PFKFB3 knockout elicited replication in either healthy β-cells or facilitated the transition of bihormonal cells to healthy β-cells with replication propensity.

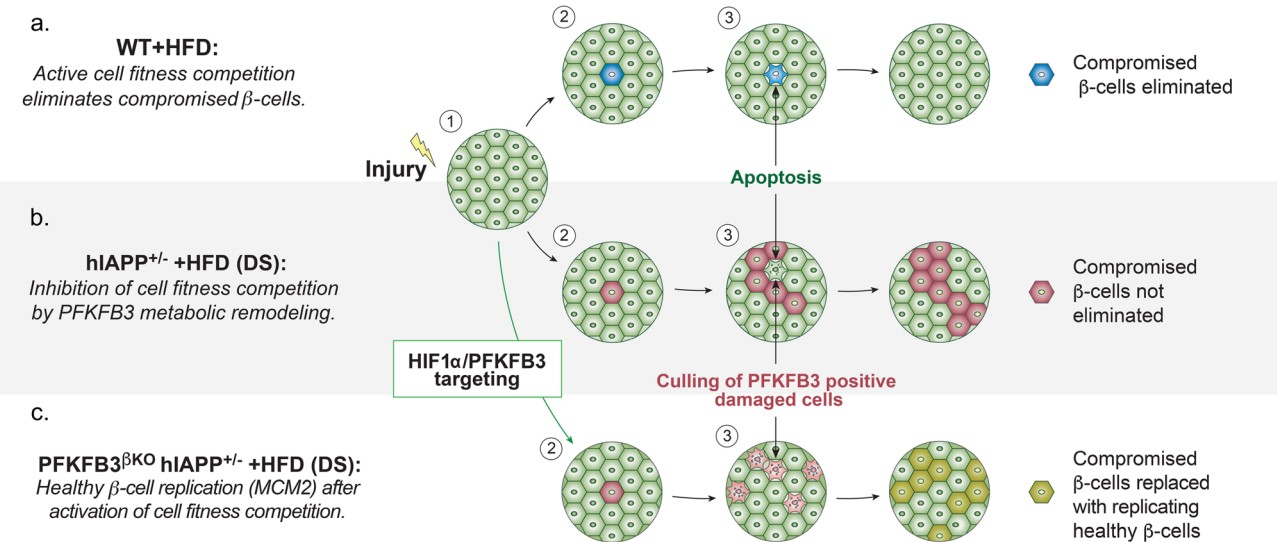

**Fig. 10 Role of β-cell fitness comparison in β-cell replenishment under stress.** An initial insult or injury can change the fitness level of some β-cells within β-cell populations and trigger cell competition. **a** After metabolic stress such as intake of an HFD in non-diabetic (ND) or wild-type (WT) mice, suboptimal cells (blue) are eliminated from the tissue by competition with healthy *β*-cells (green), which replicate to regenerate the lost tissue. **b** In a cell competition in which injury is sustained (T2D) or in PFKFB3^WT DS mice, injured β-cells (pink) survive inspite of reduced fitness and cannot be purged from the tissue because of metabolic remodelling by the HIF1α–PFKFB3 pathway. These injured β-cells may impede healthy β-cell replenishment (replication). **c** Under conditions described in (**b**), the targeting of the pro-survival PFKFB3 leads to activation of cell competition and elimination of suboptimal (damaged) β-cells (light pink). Elimination of suboptimal (damaged) β-cells leads to replication of the remaining healthy β-cells (MCM2-positive cells, dark green).

PFKFB3 knockout leads to restoration of β-cell replication comparable to WT mice independent of expression levels of HIF1α that have varied at high or low diabetogenic stress. These results indicated that the increment in replication was contributed by healthy β-cells and was conceivably facilitated by the excelled loss of β-cells with injury.

Based on our cumulative evidence, we propose that β-cell regeneration relies on the PFKFB3-dependent activation of β-cell competition and competitive culling of compromised β-cells under diabetogenic stress (Fig. 10). In PFKFB3^βKO DS mice (high DS), the increase in β-/α-cell ratio and the reduction in glucagon levels in comparison with PFKFB3^WT DS may have accounted for the observed trend of higher insulin sensitivity[36–38] at high DS. Altogether, this evidence further emphasises the functional dichotomy that the two master metabolic regulators convey in β-cell regeneration during diabetogenic stress.

In both PFKFB3^βKO DS under low diabetogenic stress and PFKFB3^βKO DS chow, restorative effect in glucose tolerance and propensity for replication resembled β-cell compensation observed during β-cell adaptation prior onset of diabetes. These results indicated that the diabetogenic effect of HIF1α and PFKFB3 on molecular level is reversible and depends on the containment of stress, and as such, it could be modified by diet.

We previously reported that the metabolic remodelling by HIF1α–PFKFB3 pathway in misfolded protein stress (hIAPP) recapitulated the consequences of HIF1α expression after conditional inactivation of von Hippel Lindau gene (*Vhl*) culminating in impaired glucose tolerance[13,39].

To investigate the role of HIF1α in the molecular basis for β-cell dysfunction, we analysed sc RNA-Seq data from humans with obese T2D and obese nondiabetics[23]. Since HIF1α is regulated mainly post-translationally with no changes in the transcript levels, we made use of the distinction of *LDHA*-positive versus negative β-cells. *LDHA* is a bona fide target-, serving as a substitute marker for HIF1α. *LDHA*-positive β-cells could not be correlated with PFKFB3 expression levels because the latter were low to undetectable (Supplementary Fig. 9). *LDHA*-positive β-cells overlapped with cluster 7 β-cells and were represented by HIF1α– (*GK, PFKFP, PDK4, HK1, HK3 and SLC2A1*), bihormonal signature (α- and β-cell identity) (*GCG, ARX and INS*) and some markers of immaturity (*ALDH1A1*). Double-insulin- and glucagon-positive cells in our mouse model resembled *LDHA*-positive or cluster-7 β-cells, possibly originating from transdifferentiation[40].

Importantly, mouse HIF1α-positive β-cells differed from the human *LDHA*-positive cells with HIF1α signature and bihormonal status, because they persisted in PFKFB3^βKO DS mice under high DS where bihormonal cells were depleted. This further highlighted the specific role that PFKFB3 and not HIF1α plays in the generation or sustenance of the bihormonal cells.

IPA used for comparison between *LDHA*-positive (cluster 7) and *LDHA*-negative (cluster 1) β-cells in both health and T2D, indicated the existence of a master upstream regulator LXR/RXR[41], known for an independent role in aerobic glycolysis in response to HFD[27,28].

Previous reports indicated that the chronic activation of LXR may contribute to β-cell dysfunction by the accumulation of free fatty acids and triglycerides[41].

In addition, the STRING analysis indicated that, while differences between clusters 1 and 7, as well as *LDHA*-negative and -positive β-cells, were well preserved in health, these differences were reduced in T2D. Preserved features of bihormonal cells relative to β-cells could constitute a basis for bihormonal cells' recognition and homoeostatic control in the context of cell fitness competition.

Cell-fitness competition is an important extrinsic cell-quality control based on the distinction of cell population with inferior versus cell population with superior ("winner") fitness characteristics. The distinction is key in triggering selective culling of the cell population with inferior fitness characteristics ("losers") and propelling the expansion of the cell population with superior fitness

characteristics ("winners"). Interestingly, proteotoxicity (similar to IAPP) was identified as the underlying cause of the "loser" status[42]. Replacement of the "losers" (injured cells) with the "winners" (healthy cells) would be in particular important in the disease to re-establish homoeostatic control over tissue quality. By implication, the reversal of cell-competitive tissue makeup such as we see in T2D (clusters 1 and 7 resemblance) would evolve in tissue failure following the accumulation of injured cells. Interestingly, obesity suppresses cell competition and with this, it can facilitate accumulation of injured cells over time[43].

In conclusion, the preservation of the β-cell mass and increase in the β-/α-ratio in the PFKFB3$^{βKO}$ DS mice stemmed cumulatively from β-cell replication in response to the depletion of injured β-cells and reduction in bihormonal cells.

While previously differentiation was discussed to restore β-cell mass, here we demonstrate how selective purification via PFKFB3 disruption can target the same bihormonal cell population[44]. One advantage of the latter strategy would rely on the compensatory hypertrophic/hyperplastic growth after selective elimination of β-cells with lower fitness. For example, in the model of Alzheimer's disease in Drosophila, activation of cell competition that purged damaged neurons was sufficient to stimulate restoration of cognitive function[44]. It is tempting to propose that PFKFB3 plays a key role in the inhibition of β-cell competition under diabetogenic stress. Discovery of β-cell replication following activation of β-cell competition in T2D could have broader implications for postmitotic and highly specialised cells, that similar to β-cells, have to mount replication under physiological constraints of tissue size and available space.

β-cell competition may also explain a negligible rate of replication in human adult β-cells. Unlike exterior organs such as skin where cells undergo high turnover with high replication being bound to high rates of cell death, in pancreas of healthy individuals, the rate of β-cell replication reflects the low rate of injury/ death. Given that replication during cell-fitness competition occurs as quid pro quo relative to cells undergoing cell death, the silent phenotype as a result will have no effect on the β-cell mass. As such, in T2D, β-cell mass does not change dramatically, but β-cell function is compromised due to accumulation of unpurged injured β-cells. The fact that the HIF1α and PFKFB3 upregulation and the loss of β-cell function happen concomitantly in pre-diabetes implicates further the inhibition of β-cell fitness competition at the early β-cell decompensation stage.

Here we demonstrate that HIF1α and PFKFB3 targeting can unlock the physiological β-cell competition under stress, providing a powerful tool to restore a functionally competent β-cell mass in T2D.

## Methods

**Animals.** We complied with all relevant ethical regulations for animal testing and research. The study received ethical approval ARC-2019-011-AM-003 by UCLA Animal Research Committee.

Homozygous hIAPP$^{+/+}$ mice on a FVB background were a gift from Dr Peter Butler's laboratory and have been previously described[45]. We have generated a β-cell-specific inducible PFKFB3-knockout mouse model (RIP-CreERT:PFKFB3$^{fl/fl}$) by crossing mice that carry the floxed *Pfkfb3* gene (JAX Laboratories) with mice that express Cre recombinase under the control of the rat insulin promoter (RIP-CreERT). We have crossed mice on a homozygous hIAPP$^{+/+}$ background with either PFKFB3$^{fl/fl}$ or RIP-CreERT mice and then crossed PFKFB3$^{fl/fl}$ hIAPP$^{+/-}$ and PFKFB3$^{fl/fl}$ RIP-CreERT mice together to generate the three experimental genotypes PFKFB3$^{fl/fl}$ hIAPP$^{-/-}$, PFKFB3$^{fl/fl}$ hIAPP$^{+/-}$ and RIP-CreERT PFKFB3$^{fl/fl}$ hIAPP$^{+/-}$. These are referred to as PFKFB3$^{WT}$ hIAPP$^{-/-}$ (hereafter WT), PFKFB3$^{WT}$ hIAPP$^{+/-}$ (PFKFB3$^{WT}$ DS) and PFKFB3$^{βKO}$ hIAPP$^{+/-}$ (PFKFB3$^{βKO}$ DS). All experimental groups were subjected to an HFD. Cre-loxP recombination of the floxed sites in *Pfkfb3* was induced by intraperitoneal injection of tamoxifen (Sigma, T5648, prepared in corn oil, Sigma C8267). All experimental groups received 250 mg of tamoxifen by kilogram of body weight every other day three times for one week. The first experimental group (high diabetogenic stress, high DS) was injected at the age of 20–27 weeks and the second group (low diabetogenic stress, low DS) at the age of 7–10 weeks. The high DS mice were given a chow diet for 10 weeks after the tamoxifen injection, and then all mice were

exposed to a HFD for another 13 weeks starting from the age of 40 weeks (HFD, Research Diets Inc, New Brunswick, NJ, USA) to induce diabetes in combination with hIAPP$^{+/-}$, since only male mice homozygous for hIAPP (hIAPP$^{+/+}$) would develop diabetes spontaneously[45,46]. The group under lower DS received HFD one day after the last tamoxifen injection and they were maintained on it for 8 weeks. Half of the PFKFB3$^{βKO}$ DS mice were switched to chow diet after being fed with an HFD and they had 16-hour restricted access to food three-times per week. Mice were maintained on a 12-hour day/night cycle at the approved mice colony facility that is run by the University of California Los Angeles' (UCLA's) Institutional Animal Care and Use Committee. At 30–37 weeks of age, all mice from the high DS group were assigned to receive the HFD (35% fat w/w or 60% calories from fat, D12492). The fat composition of the HFD was 32.2% saturated, 35.9% monounsaturated and 31.9% polyunsaturated fats. The mice had *ad libitum* access to food and water for the duration of the study. Body weights and fasting blood-glucose levels were assessed weekly, and additional measurements were made on days that included glucose- and insulin-tolerance tests.

**Insulin- and glucose tolerance tests.** An intraperitoneal glucose-tolerance test (IP-GTT) was performed at 9 and 12 weeks after the start of the HFD (19 and 22 weeks after the tamoxifen injection) for high DS mice and at 4-, 5- and 6 weeks after start of the HFD for lower DS mice. Mice were fasted overnight in a clean cage and with access to water before the tests. Tail vein blood glucose was collected before and 15, 30, 60, 90 and 120 minutes after a 20% glucose bolus injection (2 g/ kg of body weight). Retro-orbital or carotid bleeding was used to collect the blood for IP-GTT prior to and 30 minutes after the glucose bolus injection, at 12 weeks and 8 weeks of HFD for high and low diabetogenic mice, respectively. For retro-orbital bleeding, mice were anaesthetised by brief exposure to isoflurane (10 seconds). Carotid bleeding was performed on the non-anesthetised mice in the lower DS group. The blood was collected in a microcentrifuge tube coated in ethylene-diamine tetraacetic acid (EDTA) buffer and the plasma was obtained by centrifuging the samples for 10 minutes (5000 RCF, 10 min, 4 °C).

An intraperitoneal insulin tolerance test (IP-ITT) was performed in conscious mice 51 and 56 days after beginning of HFD, for high and low diabetogenic stress, respectively. The mice were fasted for six hours prior to the test. To measure glucose levels, tail-vein blood was collected prior to and 20, 40 and 60 minutes after administration of insulin (0.75 IU/kg of body weight) (Lilly insulin Lispro, LLC, Indianapolis, USA).

**Glucose and insulin assays.** Fasted blood glucose was measured weekly after overnight fasting for 18 hours. The fasting regimen involved changes of cages and bedding and withdrawal of food while water was provided *ad libitum*. The blood-glucose level was measured in tail-drawn blood through use of a freestyle blood-glucose metre (Abbott Diabetes Care Inc, Alameda, CA, USA). When the blood-glucose level exceeded the detection range of the blood glucose metre, plasma glucose was determined through use of the glucose-oxidase method and analysis with a YSI 2300 STAT PLUS glucose and l-lactate analyser. The coefficient for conversion of plasma- to whole-blood glucose values was derived from 13 WT mice and was used for generating results for the IP-GTT at 12 weeks after exposure to HFD (Supplementary Table 8).

Levels of insulin, C-peptide and glucagon in plasma were determined via use of ultrasensitive enzyme-linked immunosorbent assay (ELISA) for mouse insulin (Mercodia 10-1247-01, Uppsala, Sweden), mouse C-peptide (Crystal Chem 90050, IL, USA) and mouse glucagon (Mercodia 10-1281-01, Uppsala, Sweden).

**Pancreas perfusion and isolation.** Mice were euthanised by cervical dislocation. A medial cut was made to open the abdomen and chest cavities. A cut of the right ventricle was followed by a poke of the left ventricle with a needle to inject 10 ml of cold phosphate-buffered saline (PBS) slowly for perfusion of the pancreas. After perfusion, the pancreas was placed in cold PBS and separated from other tissues, including the surrounding fat. The pancreas was then weighed after the excess PBS had been absorbed with tissue.

**Histological assessments.** After excision of smaller pieces, the pancreas was fixed in 4% paraformaldehyde (Electron Microscopy Sciences 19202, Hatfield, PA, USA) overnight at 4 °C. It was paraffin-embedded and sectioned into 4μm thicknesses. For β-cell area, peroxidase and haematoxylin staining were performed on deparaffinised sections that were sequentially incubated with the following: rabbit anti-insulin antibody (Cell Signalling Technology C27C9, Danvers, MA, USA, 1:400); F(ab')$_2$ conjugate with Biotin-SP (Jackson ImmunoResearch 711-066-152, West Grove, PA, USA, 1:100 for IHC); the VECTASTAIN ABC kit (HRP) (Vector Laboratories PK-4000, Burlingame, CA, USA); the DAB substrate kit (HRP) (Vector Laboratories SK-4100, Burlingame, CA, USA) and Harris haematoxylin. The sections were mounted with Permount (Fisher SP15-100, Hampton, NH, USA). Morphometric analyses were performed using Image-Pro Plus 5.1 software on the Olympus IX70 inverted tissue-culture microscope (Olympus, Center Valley, PA, USA). Imaging and data analysis were performed by two observers in a blinded fashion for each section of the experimental mouse genotype.

The islet edges were manually circumscribed using a multichannel image. Insulin- and haematoxylin-positive areas were determined for each islet through

application of pixel thresholding. The β-cell area was then calculated as a percentage of the total as insulin-positive areas/haematoxylin-positive areas multiplied by 100.

Immunofluorescence analysis was performed in Openlab 5.5.0 software on the Leica DM6000 B research microscope. The following antibodies were used: rabbit anti-PFKFB3 (Abcam ab181861, Cambridge, MA, USA, 1:100); mouse anti-minichromosome maintenance-2 protein (MCM2) (BD Transduction Laboratories 610700, San Diego, CA, USA, 1:100); rabbit anti-cleaved caspase-3 (Cell Signalling Technology 9664 S, Danvers, MA, USA, 1:400); guinea pig anti-insulin (Abcam ab195956, Cambridge, MA, USA, 1:400); mouse anti-glucagon (Sigma-Aldrich G2654, St. Louis, MO, USA, 1:1000); mouse anti-c-Myc (Santa Cruz Biotechnology Inc 9E10 sc-40, Dallas, Texas, USA, 1:100); mouse anti-HIF1α (Novus Biologicals NB100-105, Centennial, CO, USA, 1:50); polyclonal rabbit anti-Mcm2 (Abcam ab441) and polyclonal rabbit anti-GADD 153 (Santa Cruz Biotechnology F-168 sc-575). The anti-PFKFB3 staining was performed as described previously[47] with some modifications. The following secondary antibodies were used: F(ab')2 conjugates with fluorescein isothiocyanate donkey anti-guinea pig immunoglobulin G (IgG) (heavy and light, H + L) (Jackson ImmunoResearch 706-096-148, West Grove, PA, USA, 1:200 for intrinsic factor (IF)); F(ab')2 conjugates with Cy3 donkey anti-rabbit IgG (H + L) (Jackson ImmunoResearch 711-166-152, West Grove, PA, USA, 1:200 for IF); F(ab')2 conjugates with Cy3 donkey anti-mouse IgG (H + L) (Jackson ImmunoResearch 711-165-151, West Grove, PA, USA, 1:200 for IF) and F(ab')2 conjugates with Alexa 647 donkey anti-mouse IgG (H + L) (Jackson ImmunoResearch 715-606-150, West Grove, PA, USA, 1:100 for IF). The In Situ Cell Death Detection Kit (Roche Diagnostics Corporation 12156792910, Indianapolis, IN, USA) was used to determine cell death by terminal deoxynucleotidyl transferase deoxyuridine triphosphate nick-end labelling (TUNEL) assay. Vectashield with 4′,6-diamidino-2-phenylindole (DAPI) (Vector Laboratories H1200, Burlingame, CA, USA) was used to mount the slides. All supporting single-channel and merged multiple-channels images for every figure or Supplementary figure are presented in Supplementary Fig. 17.

**Single-cell RNA-sequencing data analysis**. The single-cell RNA-seq dataset was obtained from the gene repository Gene Expression Omnibus (GEO, accession number GSE124742), in which healthy and T2D cells were used. To identify different cell types and to find signature genes for each cell type, the R package Seurat (version 3.1.2) was used to analyse the expression matrix. Cells in which fewer than 100 genes and 500 unique molecular identifiers (UMIs) were detected were removed from further analysis. The Seurat function NormalizeData was used to normalise the raw counts. Variable genes were identified using the FindVariableGenes function. The Seurat ScaleData function was used to scale and centre expression values in the dataset for dimensional reduction. Default parameters were used in the Seurat functions above. Principal-component analysis (PCA) and uniform manifold approximation and projection (UMAP) were used to reduce the dimensions of the data, and the first two dimensions were used in plots. The FindClusters function was used later to cluster the cells. The FindAllMarkers function was used to determine the marker genes for each cluster, and these then were used to define the cell types. Differential-expression analysis between two groups of cells was carried out using the FindMarkers function. The Wilcoxon rank-sum test was performed in the differential analysis, and the Benjamini–Hochberg procedure was applied to adjust the false-discovery rate.

**Statistics and reproducibility**. All the single values for each measurement used for generation of the graphs are presented in Supplementary Data File 6. Data are presented as errors of the means (standard error, SEM) for the number of mice indicated in the figure legends. For the IP-GTT and IP-ITT, areas under the curve (AUC) for glucose, insulin, C-peptide and glucagon were calculated through use of the trapezoidal rule[48]. Mean data were compared between groups by one-way analysis of variance (ANOVA) followed by Tukey's post-hoc test. P-values of less than 0.05 were considered significant.

**Reporting summary**. Further information on research design is available in the Nature Research Reporting Summary linked to this article.

## Data availability
The datasets generated during and/or analysed during the current study are available from the corresponding author on reasonable request.

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

## Acknowledgements

This work was supported by funding from the Larry Hillblom Foundation (Start-up Grant #2017-D-002-SUP) and Sponsored Research Agreement 2021-0206 between UCLA and Metanoia Bio Inc to STT. J.M. was supported by the Department of Endocrinology, Union Hospital of Tongji Medical College Huazhong University of Science and Technology, Wuhan, Hubei, China. We are very grateful to Madeline Rosenberger and Dr. Tatyana Gurlo for their help and advice regarding the animal breeding and technical issues in the experiments. We thank Dr. Andrea Mattern and the Division of Laboratory Medicine at UCLA for their exceptional support in the mice experiments. We also thank Dr. Lulu Chen and Dr. Tianshu Zeng for their critical reading of the paper.

## Author contributions

J.M. and B.S. performed all the in vivo experiments, analysis and prepared the data, F.M. and M.P. performed all the scRNA-Seq data quality control, analysis and related figures, S.M. designed some experiments and critically assessed and wrote part of the paper, O.G. designed some experiments and critically assessed parts of the data, S.T. conceptualised the idea, designed the experiments, analysed and assessed the data and wrote the paper.

## Competing interests

Slavica Tudzarova declares a financial competing interest in Metanoia Bio Inc. All other authors declare no competing interests.
