## [Peer Review File · Communications Biology]

Reviewers' comments:

Reviewer #1 (Remarks to the Author):

In this manuscript by Min et al, the authors created conditional knockout of PFKFB3 in beta cells with overexpression of hIAPP and high fat diet (PFKFB3 β KO DS mice). The authors performed insulin/glucose tolerance test and examined the pancreatic tissue sections via immunofluorescent with various markers. The authors stated that PFKFB3 appear to play a role in cell fitness to allow for the out competition of healthy beta cells while eliminating damaged beta cells. While this concept is intriguing, I have some questions/comments on the interpretations of the experimental data. Specifically:

1. To enhance the argument that healthy beta cells contribute to the increased replication in the PFKFB3 β KO DS mice, have the authors tried to co-label c-Myc and MCM2 in the beta cells? Is there a depletion of MCM2 positivity in the c-Myc+ cells?
2. For the differential expression analysis presented in Figure 6, it may enhance interpretability if the authors summarize similarity and differences between different comparison groups using venn diagrams.
3. The authors argue that PFKFB3 β KO DS mice specifically cleared the bihormonal cells. However, the experimental results are endpoint staining indicating fewer bihormonal cells with the other two genotypes. Could the author co-label INS, GCG with caspase-3/tunnel and demonstrate that the preference "culling of bihormonal cells" in the PFKFB3 β KO DS mice? Time course data would also help to substantiate the conclusion and confirm that the PFKFB3 β KO DS mice did have more bihormonal cells to start with.
4. PFKFB3 β KO DS mice have increased proliferation of healthy beta cells, reduced number of stressed beta cells, and comparable % of c-myc and HIF1 α + beta cells to wildtype, but why these mice were the most glucose intolerant? In light of the physiological phenotype of these PFKFB3 β KO DS, the authors' statement of the potential therapeutical value of selectively disrupt PFKFB3 seems to be unjustified.

Minor points:

1. Genotype of Animals under materials and methods do not correspond to the annotation.
2. Single-cell analysis. What is going on for alpha and acinar cells that seem to be high in general for LDHA? I understand this is not the main point of the paper but maybe worthwhile looking into.

Reviewer #2 (Remarks to the Author):

The manuscript by Jie Min, et al., entitled " PFKFB3 depletion activates beta cell replication by cell competitive culling of compromised beta cells under stress" describes the role of PFKFB3 in pancreatic beta cell function and mass. The authors elegantly show that beta-cell specific deletion of PFKFB3, in a HFD-IAPP background, leads to removal of bihormonal, i.e. insulin+/glucagon+ beta cells, increases beta cell proliferation and replenishment of beta cells. These data strongly suggest that depleting this enzyme could lead to beta cell replenishment during type 2 diabetes. They also observed that, in these KO models, glucose tolerance and in vivo insulin secretion in response to glucose were comparable to WT, associated to high number of HIF1 α positive cells, suggesting that a compensatory mechanism leading to alternate HIF1 α program could maintain impairment of glucose homeostasis in this diabetic background. Using several mouse models and data mining of single cell analysis, the authors suggest an important role of PFKFB3 in beta cell impairment during T2D.

In figure 1, HIF1 α levels remained increased in PFKFB3 beta KO. The authors mention that "this results indicated that PFKFB3 KO triggered a compensatory HIF1 α expression in response to stress". This statement is, to the reviewer's opinion, not demonstrated by the data. Alternatively, it can also be independent of PFKFB3, and/or only related to the IAPP/HFD background. Concerning the metabolic phenotype of these mice, it is not clear why the authors presented fasting insulin/beta-cell mass. This is not commonly presented. The insulin levels during IPGTT are

missing. This is probably more informative and directly related to in vivo beta cell function.

The number of mice used is limited also, and should be increased to get more statistical power.

The single cell analysis is confusing. The authors did not mention where to which cluster PFKFB3 RNA belongs, if it is detected. Then the conclusion related to the data mining of scRNA seq are interesting, but it is difficult to directly link to PFKFB3 KO mice results. We know that loss of identity appears during T2D, but whether this is related to PFKFB3 remains unknown, and at least, scRNA seq does not prove anything here. The reviewer does not see the link between scRNA seq data presented here, and the scope of the paper, that is to demonstrate that PFKFB3 beta cell specific deletion increases beta cell proliferation. The fact that bihormonal cells exist in T2D human beta cells and PFKFB3 pancreas does not mean that it is directly related. This should be proven by doing scRNA seq in WT and KO mice, but this is probably beyond the scope of this paper. I thus suggest to write the scRNA seq part to make it more comprehensible for the reader.

In the material & methods section, it is mentioned that « Retro-orbital bleeding was used to collect the blood for the second IP-GTT prior to and 30 minutes after glucose bolus injection. The mice were anaesthetised by brief exposure to isoflurane (10 seconds). » This is not common to anaesthetize mice during an IP-GTT, since isoflurane may affect insulinemia and glycemia could happen. Can you comment on that?

In addition, it is mentioned that « When blood glucose exceeded the detection range of the blood glucose meter, plasma glucose was determined using the glucose oxidase method and analyzed with YSI 2300 STAT PLUS Glucose and L-Lactate Analyzer ». It may be difficult to interpret the results when two different methods are used to measure glycemia. Can you also comment on that?

Minor

The genetic background of the mice used in this study should be mentioned, as well as the tamoxifen injection protocol (dose, number of IP injections) and the reference of the HFD.

Reviewer #3 (Remarks to the Author):

In this present manuscript, the authors used mice with beta-cell specific ablation of Pfkfb3 gene on hIAPP+/- background and a high-fat diet (PFKFB3 beta KO DS mice) to investigate the role of PFKFB3 and HIF1alpha, since PFKFB3 is a target of HIF1alpha, upon beta-cell regeneration process. Interestingly, they observed that PFKFB3 beta-cell ablation, under the diabetogenic stress (induced by hIAPP and high-fat diet), led to a compensatory beta cell replication after selective elimination of damaged beta cell following the rules of cell competition. This result is very interesting and may contribute to provide a novel strategy to maintain beta cell mass in type 2 diabetes. However, it is important to be aware that although insulin sensitivity improved in PFKFB3 beta KO DS mice, glucose intolerance seems to impair in these mice. The authors linked this glucose intolerance to the HIF1alpha compensatory response that occurs in PFKFB3 beta KO DS mice, but this matter needs more investigation.

The manuscript is well written. The methods were described with details, but the statistical analysis needs to be revised. Please, see the specifics comments below.

Specifics comments:

1. The title in the manuscript needs to be correct.
2. Please, provide the amount of glucose bolus injected during the IP-GTT.
3. In the methods section ("Insulin and glucose tolerance tests"), the authors only describe the glucose tolerance test (IP-GTT). Insulin tolerance test (IP-ITT) is described in "Glucose and insulin assays".
4. Why IP-GTT was assess after 19 and 22 week and IP-ITT was assess only after 19 weeks from tamoxifen injection, as described in the methods? Why the authors did not show the results from 22 weeks?

5. "Data are presented as an error of the mean (standard error, SEM) for the number of mice indicated.". Please, added "indicated in the figure's legends".
6. The "statistical analyses" section should be better described. The authors could explain with more detail de calculation of AUC using the trapezoidal rule or cite a reference that explain it. Also, it was described that Student's t-test was used to compare the mean data between groups, but all experiments presented in the manuscript have three groups. ANOVA one-way followed by a post-hoc test should be used to compare three groups.
7. Please, provide the fasting insulin and c-peptide levels as you provided fasting glucagon levels (Figure 2F).
8. Is it correct using "fasting insulin/beta-cell mass" to refer to "fasting insulin/c-peptide"? The authors should know that lower insulin/c-peptide ratio may suggest an increase in the insulin clearance (Piccinini and Bergman. The Measurement of Insulin Clearance. Diabetes Care 2020 Sep; 43(9): 2296-2302.). Thus, PFKFB3 beta KO DS mice seem to display an increase in insulin clearance process, which could impact plasma insulin levels.
9. Based on the results from Figure 2, the authors suggest impaired insulin secretion in the PFKFB beta KO DS mice. However, the results displayed in this figure did not support this suggestion. The authors should, at least, show the c-peptide levels during the IP-GTT, or evaluating insulin secretion with more directly methods (for example, glucose stimulated insulin secretion in isolated islets).

POINT-BY-POINT ANSWERS TO REVIEWERS COMMENTS AND CONCERNS

Thank you for reviewing our manuscript entitled "PFKFB3 DEPLETION ACTIVATES β -CELL REPLICATION BY CELL COMPETITIVE CULLING OF COMPROMISED β -CELLS UNDER STRESS" -

Here are our point-by-point answers to reviewers' comments and questions:

Reviewer #1

Thank you very much for the constructive and relevant questions that are critical to the improvement of the manuscript and to leverage the knowledge gained through the findings in our work.

Q1. To enhance the argument that healthy beta cells contribute to the increased replication in the PFKFB3 β KO DS mice, have the authors tried to co-label c-Myc and MCM2 in the beta cells? Is there a depletion of MCM2 positivity in the c-Myc+ cells?

A1. Thank you for this excellent suggestion, which adds clarity regarding the status of replicating MCM2-positive β -cells in PFKFB3 β KO DS mice under ongoing stress. We co-labelled MCM2 with c-Myc and insulin in the three experimental groups as suggested. To identify a potential overlap, we included β -cells with low expression levels of MCM2. Overlap between c-Myc and MCM2-positive β -cells was observed in less than 0.6% of all β -cells in each experimental group and no significant difference was observed between the groups (please see the figure below and Supplementary Fig. 16). The lower panel in the images shows the rare occurrence of MCM2 and c-Myc co-labelling, which indicated that the outcome was not due to a technical flaw. Our data indicate that MCM2 positivity is depleted in cytoplasmic c-Myc-positive β -cells. Cytoplasmic c-Myc corresponds to the c-Myc truncation, which is a consequence of calpain processing of c-Myc during misfolded stress (incurred by hIAPP). These data confirm that MCM2-positive cells are free of hIAPP-related stress and undergo replication to an extent comparable with the WT controls. In marked manuscript text please refer to line 483-485 highlighted in grey and Figure Legend for Supplementary Figure 16 and below.

Q2. For the differential expression analysis presented in Figure 6, it may enhance interpretability if the authors summarize similarity and differences between different comparison groups using Venn diagrams.

A2. That was a great idea and application of it helped us to reduce the complexity of the supplementary figures as we summarised the similarities and differences between different groups through use of Venn diagrams. As a result, the statement regarding the striking reduction in the size of the subset of differentially expressed genes in type 2 diabetes (T2D) compared with the size of this subset in healthy non-diabetics was obviated.

Please see the figure below (Figure 6e,f), which confirms the incorporation of the Venn diagrams and highlight the comments that have been incorporated in the revised manuscript. **In marked manuscript text please refer to line 399-401 highlighted in grey and Figure Legend for Figure 6e,f.** Here (see below), **e** shows the distribution of shared and distinctive gene markers in Cluster 7 versus Cluster 1, while **f** shows the shared and distinctive gene markers in LDHA-positive versus LDHA-negative β -cells between non-diabetics (ND) and type-2 diabetes (T2D).

Q3. The authors argue that PFKFB3 β KO DS mice specifically cleared the bihormonal cells. However, the experimental results are endpoint staining indicating fewer bihormonal cells with the other two genotypes. Could the author co-label INS, GCG with caspase-3/tunnel and demonstrate that the preference “culling of bihormonal cells” in the PFKFB3 β KO DS mice? Time course data would also help to substantiate the conclusion and confirm that the PFKFB3 β KO DS mice did have more bihormonal cells to start with.

A3. First, we repeated the mouse experiment with high fat exposure for 8 weeks (see d, e, f in the figure below, Figure 7) and reduced the diabetogenic stress (DS) to compare the results at early (eight-week) and late (13-week) time points (a, b, c in the figure below). PFKFB3 WT DS mice exposed to misfolded protein and a high-fat diet had increased numbers of bihormonal cells at both time points at comparable levels, as shown in the figure. It has been reported in the literature that bihormonal cells may represent a transient pool that can be mobilised to complement either β - or α -cells under stress (Figure 7a-f).

The PFKFB3 β KO DS group had the lowest number of bihormonal cells at both time points, which indicated that they were inversely correlated with PFKFB3 expression. As recommended by the reviewer, **we performed insulin, glucagon and caspase-3 co-labelling and compared the**

frequencies of insulin and glucagon co-expressing cells with that of cleaved caspase-3 positive cells. We observed no cleaved caspase-3 co-labelling with the bihormonal cells in any experimental group, including in PFKFB3 β KO DS mice (Figure 7g). These results imply that PFKFB3 expression is necessary for the generation of bihormonal cells during adaptation to stress, since there was no progressive pro-apoptotic labelling or loss of these cells in the PFKFB3 β KO DS group over the early- and late time points that were studied.

In marked manuscript text please refer to line 425-428 highlighted in grey and Figure Legend for Figure 7 including 7g. Please see the results about bihormonal cells below.

Q4. PFKFB3 β KO DS mice have increased proliferation of healthy beta cells, reduced number of stressed beta cells, and comparable % of c-myc and HIF1 α + beta cells to wildtype, but why these mice were the most glucose intolerant? In light of the physiological phenotype of these PFKFB3 β KO DS, the authors' statement of the potential therapeutical value of selectively disrupt PFKFB3 seems to be unjustified.

A4. We apologise for the confusion that was caused to the reviewer – we did not explain clearly to the reader that PFKFB3 β KO DS mice under high diabetogenic stress had reduced damage as measured by cytoplasmic c-Myc (Figure 9e) while contained significantly increased numbers of HIF1 α + β -cells when compared to the WT and similarly increased numbers of HIF1 α + β -cells when compared with the PFKFB3 WT DS mice. HIF1 α expression levels in the PFKFB3 β KO DS mice were similar to those observed in the PFKFB3 WT DS group under high DS (Figure 1d).

Further, we identified this HIF1 α + β -cell subpopulation via the surrogate marker LDHA (a bona fide target of HIF1 α) after scRNA Seq of pancreases from humans with T2D and from non-diabetics with high BMI. We found that LDHA (disallowed gene)-positive β -cells showed co-expression of insulin (pro- β -cell marker), glucagon and ARX (pro-alpha cell markers) and some markers of immaturity (ALDH1A1). Therefore, they mimicked the bihormonal β -cells that were observed in the presence of PFKFB3 in mice under diabetogenic stress (DS).

To confirm that HIF1 α expression is linked to β -cell dysfunction in PFKFB3 β KO DS under high DS (as implied from scRNA Seq data above), we undertook a dual strategy. First, we repeated the experiment in mice that had been exposed to shorter periods of DS (7-10 weeks of age and 8 weeks of exposure to high-fat diet) and introduced the fourth group of PFKFB3 β KO DS mice, which had been taken off a high-fat diet five weeks after the diabetogenic phenotype occurred and fed with a chow diet for 16 hours per day 3 times a week (presented in green). The results are shown in the figure below (Figure 3, see below).

We found that cytoplasmic c-Myc (calpain activation) was not detectable at lower DS and instead that ER stress marker CHOP was elevated in the PFKFB3 WT DS group but was abated in PFKFB3 βKO DS. These results indicated that HIF1α expression at higher DS correlated with misfolded protein stress (calpain activation). In marked manuscript text please refer to line 511-518 highlighted in grey and Figure Legend for Supplementary Figure 15b,c. Please see below.

Supplementary Figure 15

Clearly, at lower levels of DS, the PFKFB3 βKO DS mice performed better metabolically than those that were exposed to higher DS and their performance was comparable with that of

WT mice in terms of the intra-peritoneal glucose tolerance test (IP-GTT). The fasting blood glucose levels of the PFKFB3 β KO DS mice were lower consistently than those of the PFKFB3 WT DS throughout the experiment from day 21 with significance at days 21, 31, 38 and 52. When a chow diet was substituted for the high-fat diet in the PFKFB3 β KO DS mice, they performed even better as measured by the IP-GTT than did the WT controls on a high-fat diet. Interestingly, this result correlated with the expression of HIF1 α , which was significantly reduced in both PFKFB3 β KO DS mice and in PFKFB3 β KO DS mice chow diet compared with PFKFB3 WT DS mice.

These results indicate that the decompensation of the metabolic improvement after PFKFB3 knockout depends on the levels of HIF1 α expression. Increased HIF1 α immunolabelling is found to correlate with poor metabolic performance and vice versa, which indicates that PFKFB3 knockout can restore the function in regenerating β -cells only in the absence or at low expression levels of HIF1 α .

Please see the figure below (Figure 4), which shows HIF1 α expression diminished in PFKFB3 β KO DS and PFKFB3 β KO DS chow under lower DS (age 7-10 weeks and eight weeks of high-fat diet). In marked manuscript text please refer to line 299-341 highlighted in grey and Figure Legend for Figure 3 and 4. Figure 4 results are copied below.

Minor points:

Q1. Genotype of Animals under materials and methods do not correspond to the annotation.

A1. Thank you for identifying the omission, we have now corrected it. In marked manuscript text please refer to line 98-109 highlighted in grey Material and Methods.

Q2. Single-cell analysis. What is going on for alpha and acinar cells that seem to be high in general for LDHA? I understand this is not the main point of the paper but maybe worthwhile looking into.

A2. Below, we present a comparison between LDHA-positive and LDHA-negative alpha (α) cells from healthy and T2D mice. We did not compare them in acinar cells because there were only four healthy acinar cells available. Although there were 29 acinar cells from T2D individuals, 28 of them were LDHA-positive, so the comparisons in the acinar cells did not seem meaningful at the available cell numbers.

Please see below the differentially expressed genes in LDHA-positive versus LDHA-negative α cells in healthy non-diabetic donors and in T2D. In marked manuscript text please refer to line 387-389 highlighted in grey Supplemental Tables 9a,b and 10.

Supplementary Table 9a - LDHA-positive_vs_LDHA-negative α cells in health

	p_val	avg_logFC	pct.1	pct.2	p_val_adj	Cluster	Gene
LDHA	8.76459545566544E-101	1.7157437503042	1	0	4.09245255611387E-96	LDHA+	LDHA
PPP1CA	9.68603102977867E-14	0.264952297271143	0.796	0.502	4.52269846873455E-09	LDHA+	PPP1CA
NOL7	3.96336211798535E-12	0.264808310511192	0.842	0.622	1.8506126737509E-07	LDHA+	NOL7
CYC1	1.30672746772977E-11	0.31023545755873	0.874	0.613	6.10150256507062E-07	LDHA+	CYC1
NUDT5	3.42311266633489E-11	0.242205273255763	0.626	0.338	1.59835399729175E-06	LDHA+	NUDT5
LEPROTL1	1.13746540541719E-10	0.288519961677659	0.633	0.378	5.31116721751451E-06	LDHA+	LEPROTL1
YIF1A	1.16484114685327E-10	0.325000794413067	0.887	0.707	5.43899276700196E-06	LDHA+	YIF1A
TMEM60	1.44209328373882E-10	0.268688167420588	0.714	0.449	6.7335661697617E-06	LDHA+	TMEM60
STOML2	2.96391842619725E-10	0.19264822129713	0.717	0.44	1.38394243074428E-05	LDHA+	STOML2
TMEM258	3.75861497895907E-10	0.166886248653889	0.961	0.796	1.75501009212536E-05	LDHA+	TMEM258
NDUFB10	4.11723505205446E-10	0.165756196797671	0.877	0.631	1.92246056285579E-05	LDHA+	NDUFB10
YIF1B	6.79221036551571E-10	0.192254600574405	0.601	0.342	3.17148678597025E-05	LDHA+	YIF1B
C19orf70	7.59188544165287E-10	0.196535815126965	0.872	0.653	3.54487906927098E-05	LDHA+	C19orf70
MRPL20	7.95848975641249E-10	0.216346542526558	0.835	0.596	3.71605762196169E-05	LDHA+	MRPL20

	p_val	avg_logFC	pct.1	pct.2	p_val_adj	Cluster	Gene
CCT7	1.17179618870465E-09	0.30003078690564	0.882	0.707	5.47146794391861E-05	LDHA+	CCT7
NME2	1.22507324023481E-09	0.153425955269147	0.872	0.649	5.72023448062841E-05	LDHA+	NME2
HSBP1	1.25152962108749E-09	0.251596691621515	0.938	0.782	5.84376725974383E-05	LDHA+	HSBP1
KIAA2013	1.29251147928771E-09	0.130680609683196	0.702	0.409	6.0351238502381E-05	LDHA+	KIAA2013
FKBP1A	1.37496167662982E-09	0.256154340680376	0.911	0.707	6.42010855668761E-05	LDHA+	FKBP1A
ILVBL	1.48403062160009E-09	0.234979118196551	0.655	0.382	6.9293841814373E-05	LDHA+	ILVBL
RER1	2.9991687810334E-09	0.254005544203033	0.685	0.453	0.000140040187892793	LDHA+	RER1
ATP5B	3.83559879963553E-09	0.28678121707048	0.99	0.924	0.000179095614751382	LDHA+	ATP5B
PSMC3	4.11850691164393E-09	0.253526431626231	0.892	0.662	0.00019230544322539	LDHA+	PSMC3
SSBP1	4.21622959466681E-09	0.251689188136933	0.865	0.684	0.000196868408463777	LDHA+	SSBP1
ABRACL	4.5150123299212E-09	0.166867715747317	0.466	0.218	0.000210819470721011	LDHA+	ABRACL
FUCA2	5.76688485482923E-09	0.302569178628102	0.788	0.556	0.000269273154526541	LDHA+	FUCA2
AURKAIP1	6.57371087246698E-09	0.198284564671381	0.894	0.649	0.000306946281768101	LDHA+	AURKAIP1
CCT3	7.75033754410081E-09	0.276778667962505	0.889	0.738	0.000361886510946699	LDHA+	CCT3
ATP5G1	7.86915976398864E-09	0.214847178477868	0.86	0.658	0.000367434676859921	LDHA+	ATP5G1
PEX16	9.21057038087791E-09	0.158703943685957	0.594	0.342	0.000430069162794332	LDHA+	PEX16
TCEAL8	1.09925800021315E-08	0.206223120486883	0.665	0.427	0.000513276538039526	LDHA+	TCEAL8
RPN1	1.1204289523367E-08	0.287762341900089	0.953	0.8	0.000523161890714575	LDHA+	RPN1
TRMT112	1.12338604553169E-08	0.226436410253632	0.943	0.791	0.000524542646240112	LDHA+	TRMT112
MRPL21	1.18408257072079E-08	0.197998857066943	0.732	0.507	0.00055288367474666	LDHA+	MRPL21
CSTB	1.26818697021201E-08	0.263434625352816	0.936	0.76	0.000592154542001093	LDHA+	CSTB
ATP5A1	1.27917468266185E-08	0.290745213399365	0.966	0.831	0.000597285034575297	LDHA+	ATP5A1
PPP5C	1.31825968996966E-08	0.185998681567869	0.584	0.351	0.000615534997037535	LDHA+	PPP5C
TUFM	1.3703668799199E-08	0.261231486763051	0.867	0.68	0.000639865407241001	LDHA+	TUFM
POP4	1.41692264875198E-08	0.222073835355576	0.584	0.356	0.000661603692381761	LDHA+	POP4

	p_val	avg_logFC	pct.1	pct.2	p_val_adj	Cluster	Gene
SYNGR2	1.44874167194957E-08	0.287078849351312	0.833	0.636	0.000676460948883411	LDHA+	SYNGR2
CD47	1.65169432906879E-08	0.394314616472542	0.788	0.636	0.000771225633072092	LDHA+	CD47
POLR2C	1.70942872979123E-08	0.158214575772198	0.704	0.476	0.000798183556801419	LDHA+	POLR2C
COPE	1.74225087975134E-08	0.210404116777519	0.948	0.791	0.000813509203282295	LDHA+	COPE
PTGES2	1.77466298975462E-08	0.232123026278414	0.663	0.427	0.000828643389806125	LDHA+	PTGES2
SMARCB1	1.79714310477204E-08	0.121079146141872	0.665	0.436	0.00083914002991121	LDHA+	SMARCB1
KARS	2.26779804278084E-08	0.27976145996818	0.717	0.498	0.00105890294011566	LDHA+	KARS
PTP4A3	2.29799187414887E-08	0.352737827202742	0.557	0.311	0.00107300134579633	LDHA+	PTP4A3
RPS4Y1	2.4009207653708E-08	0.27171261393284	0.495	0.262	0.00112106193297459	LDHA+	RPS4Y1
AUP1	2.42511497923323E-08	0.213473710432069	0.798	0.547	0.00113235893725337	LDHA+	AUP1
TSR3	2.66659250511137E-08	0.202772491541439	0.685	0.431	0.00124511203841165	LDHA+	TSR3
SQSTM1	3.05676910726944E-08	0.39268992846221	0.995	0.978	0.00142729719925732	LDHA+	SQSTM1
PSMA1	3.07906369285433E-08	0.204345964506462	0.901	0.716	0.00143770721010447	LDHA+	PSMA1
LSM 2.00	3.22703128355738E-08	0.120557291002549	0.53	0.307	0.00150679771723145	LDHA+	LSM 2.00
FARSA	3.42926827331146E-08	0.229528475376197	0.645	0.418	0.00160122823485732	LDHA+	FARSA
ELOVL1	3.59759994650774E-08	0.221970421102745	0.766	0.596	0.00167982734302286	LDHA+	ELOVL1
BANF1	3.92240648681503E-08	0.144152955692021	0.791	0.56	0.00183148926088854	LDHA+	BANF1
GNAZ	4.05356085634938E-08	0.133713133576944	0.628	0.391	0.00189272917065521	LDHA+	GNAZ
RPL27	4.10418658287154E-08	0.179717688648918	0.98	0.933	0.00191636784114021	LDHA+	RPL27
MLF 2.00	4.71432845628724E-08	0.211651310079069	0.889	0.698	0.0022012613860942	LDHA+	MLF 2.00
RNH1	4.86681009404821E-08	0.235942327100843	0.867	0.671	0.00227245963721393	LDHA+	RNH1
CYB5A	4.87102302071039E-08	0.198469620495023	0.581	0.351	0.0022744267790603	LDHA+	CYB5A
RALA	4.95902920406996E-08	0.184486688330836	0.724	0.489	0.00231551950625638	LDHA+	RALA
TPI1	5.4340210062246E-08	0.237048030992978	0.978	0.862	0.00253730742843645	LDHA+	TPI1
MRPL42	5.50316401141917E-08	0.264980253122948	0.714	0.489	0.00256959237185195	LDHA+	MRPL42

	p_val	avg_logFC	pct.1	pct.2	p_val_adj	Cluster	Gene
CRYZ	5.62934582523783E-08	0.321577785266361	0.66	0.462	0.0026285104461783	LDHA+	CRYZ
SLC17A5	5.94555191510555E-08	0.219349830811661	0.66	0.436	0.00277615655572023	LDHA+	SLC17A5
AIP	6.25244790317764E-08	0.200996843463741	0.631	0.409	0.00291945549943073	LDHA+	AIP
EIF3K	6.31359576097967E-08	0.201321926387096	0.897	0.689	0.00294800726867424	LDHA+	EIF3K
FAM131C	6.66596819804492E-08	0.106307834394294	0.68	0.427	0.00311254053071311	LDHA+	FAM131C
GABARAP	6.7316790003804E-08	0.143741173089634	0.973	0.871	0.00314322287564762	LDHA+	GABARAP
THOC7	7.02080289499326E-08	0.174123700499404	0.7	0.502	0.0032782234957592	LDHA+	THOC7
CCT5	7.8126158799053E-08	0.218425540879562	0.842	0.627	0.00364794473280418	LDHA+	CCT5
DCTPP1	7.83459103643146E-08	0.108543593683635	0.502	0.249	0.00365820559264094	LDHA+	DCTPP1
PPCS	7.91991140046792E-08	0.214735094861298	0.645	0.387	0.00369804423022048	LDHA+	PPCS
ETFA	8.47079664855246E-08	0.211737303726306	0.69	0.449	0.0039552690791086	LDHA+	ETFA
PLEKHJ1	8.51385327043721E-08	0.185199236958617	0.638	0.413	0.00397537350756525	LDHA+	PLEKHJ1
ZNHIT1	8.90180767398436E-08	0.16587968176718	0.81	0.618	0.00415652105721352	LDHA+	ZNHIT1
TIMM13	9.36380298069824E-08	0.151369915344856	0.793	0.578	0.00437224052577743	LDHA+	TIMM13
KRTCAP2	9.99880163642094E-08	0.122347879292812	0.933	0.751	0.00466874044809403	LDHA+	KRTCAP2
PPP1R11	1.05631999699278E-07	0.167964859487207	0.682	0.44	0.0049322749619584	LDHA+	PPP1R11
PSMD8	1.1039929441034E-07	0.236962230805027	0.958	0.844	0.00515487425390199	LDHA+	PSMD8
PPP6R2	1.11860974646481E-07	0.162669178272174	0.724	0.529	0.00522312448916815	LDHA+	PPP6R2
PRDX5	1.22914346542607E-07	0.205659550760895	0.963	0.836	0.00573923958311397	LDHA+	PRDX5
EXT2	1.23627740707624E-07	0.177245872576926	0.458	0.24	0.00577255009686111	LDHA+	EXT2
SLC25A4	1.24549553115053E-07	0.218211312687938	0.835	0.604	0.00581559228360117	LDHA+	SLC25A4
LMAN1	1.31904320697318E-07	0.233626844302103	0.825	0.662	0.00615900844631987	LDHA+	LMAN1
TMED1	1.38741167249631E-07	0.125581874544989	0.505	0.271	0.00647824132238701	LDHA+	TMED1
BLVRB	1.39739202242316E-07	0.236852714770511	0.65	0.471	0.00652484257030046	LDHA+	BLVRB
COX17	1.41924097960114E-07	0.150289129705018	0.941	0.809	0.00662686190605161	LDHA+	COX17

	p_val	avg_logFC	pct.1	pct.2	p_val_adj	Cluster	Gene
ANAPC11	1.44497503790875E-07	0.166742721788818	0.936	0.751	0.00674702194450732	LDHA+	ANAPC11
ARPC3	1.45119237500478E-07	0.250493995633112	0.936	0.791	0.00677605255660981	LDHA+	ARPC3
MRPL3	1.53064593534692E-07	0.222390579780196	0.709	0.484	0.00714704506591535	LDHA+	MRPL3
PAPSS2	1.54117510204922E-07	0.353972211802406	0.773	0.604	0.00719620890399842	LDHA+	PAPSS2
AGTRAP	1.6165866422692E-07	0.134660140382606	0.493	0.258	0.00754832800874755	LDHA+	AGTRAP
VPS25	1.7438017303338E-07	0.209388759798612	0.675	0.44	0.00814233341944759	LDHA+	VPS25
PPIE	1.76927683626727E-07	0.186528145882437	0.599	0.378	0.00826128433158277	LDHA+	PPIE
TOMM22	1.9836360377322E-07	0.198964062981582	0.702	0.498	0.00926219175098298	LDHA+	TOMM22
ZFAND2B	2.03360393892143E-07	0.147270738987041	0.709	0.453	0.00949550687200583	LDHA+	ZFAND2B
RHOC	2.08011360237667E-07	0.142392582910518	0.675	0.436	0.00971267444357737	LDHA+	RHOC
ALKBH7	2.12824972610202E-07	0.137716623377643	0.64	0.404	0.00993743644608818	LDHA+	ALKBH7
TBCB	2.14395076561377E-07	0.225581720159066	0.815	0.609	0.0100107493098804	LDHA+	TBCB
OARD1	2.19597288756432E-07	0.192744243159361	0.515	0.302	0.0102536562039041	LDHA+	OARD1
LMAN2	2.40650695046092E-07	0.215487991726381	0.862	0.658	0.0112367029037872	LDHA+	LMAN2
ABCG2	2.46001443656437E-07	0.147183042941915	0.214	0.058	0.01148654540865	LDHA+	ABCG2
COPZ1	2.49940823476097E-07	0.258206942135226	0.899	0.707	0.0116704868705694	LDHA+	COPZ1
PLK2	2.52342046323181E-07	0.304858758179601	0.7	0.489	0.0117826071689683	LDHA+	PLK2
CD36	2.57744272833652E-07	0.367838174475829	0.525	0.32	0.0120348533314217	LDHA+	CD36
PHB2	2.61622753974771E-07	0.19156034983391	0.729	0.529	0.012215951251344	LDHA+	PHB2
ECHS1	2.66207426311065E-07	0.233092483581149	0.773	0.569	0.0124300233567426	LDHA+	ECHS1
PPIA	2.76041755532929E-07	0.198543318997948	0.988	0.924	0.012889217691099	LDHA+	PPIA
LIMCH1	2.82960188924784E-07	0.287852727442989	0.877	0.724	0.013212260101465	LDHA+	LIMCH1
PCMT1	2.83141287493999E-07	0.258309883305034	0.911	0.68	0.0132207161369573	LDHA+	PCMT1
GAPDH	2.87473407342627E-07	0.235234179840152	1	0.991	0.0134229958090493	LDHA+	GAPDH
ANAPC13	2.91868317520184E-07	0.22727557243054	0.739	0.516	0.0136282073499699	LDHA+	ANAPC13

	p_val	avg_logFC	pct.1	pct.2	p_val_adj	Cluster	Gene
NDUFC2	2.98724871876826E-07	0.181369692358895	0.958	0.822	0.0139483604425447	LDHA+	NDUFC2
PLP2	3.16229933846779E-07	0.225960877628337	0.938	0.831	0.0147657243011077	LDHA+	PLP2
NT5C	3.18088117402076E-07	0.117621078635047	0.53	0.302	0.0148524884658551	LDHA+	NT5C
CHPF	3.24600482790468E-07	0.199948625733405	0.857	0.644	0.0151565703429353	LDHA+	CHPF
ASNA1	3.28824222577839E-07	0.143400926504696	0.764	0.578	0.0153537894248271	LDHA+	ASNA1
ALG1	3.63619771854045E-07	0.12818825657007	0.478	0.258	0.0169784980071809	LDHA+	ALG1
LINC00261	3.71718474742854E-07	0.118257094742589	0.468	0.253	0.0173566507411681	LDHA+	LINC00261
TUBB	3.74409098170219E-07	0.29288265676203	0.906	0.751	0.017482284020862	LDHA+	TUBB
MRPS34	3.99967554341245E-07	0.168771570495335	0.749	0.556	0.0186756850148558	LDHA+	MRPS34
PRMT2	4.19680209290169E-07	0.157398517032779	0.828	0.644	0.0195961280123859	LDHA+	PRMT2
NDUFS4	4.23105435269141E-07	0.199577146218242	0.732	0.547	0.019756062089022	LDHA+	NDUFS4
ENSA	4.56234428162607E-07	0.132125132186579	0.889	0.698	0.0213029541541966	LDHA+	ENSA
SLC9A3R1	4.65149006278104E-07	0.195751261688115	0.741	0.52	0.0217192025501435	LDHA+	SLC9A3R1
AP2M1	4.75313334123055E-07	0.23470875379647	0.897	0.68	0.0221938055102078	LDHA+	AP2M1
NIPA2	4.76779353315692E-07	0.204828316008247	0.599	0.369	0.0222622583443696	LDHA+	NIPA2
ATPIF1	4.89580434076881E-07	0.126227553172447	0.909	0.729	0.0228599792083518	LDHA+	ATPIF1
TMEM167A	4.92387897627856E-07	0.221708185731614	0.749	0.542	0.0229910681039375	LDHA+	TMEM167A
HTATSF1	4.93243652634951E-07	0.168862305980265	0.717	0.516	0.0230310258724838	LDHA+	HTATSF1
C19orf24	4.94282529041548E-07	0.109445627378867	0.717	0.52	0.023079534128537	LDHA+	C19orf24
SPG21	4.95485745631475E-07	0.298877803116473	0.709	0.533	0.0231357159207705	LDHA+	SPG21
CRB3	5.14780096273503E-07	0.144844510840311	0.586	0.36	0.0240366270352987	LDHA+	CRB3
RALY	5.1618635691849E-07	0.163114237985005	0.788	0.573	0.0241022895635951	LDHA+	RALY
MAPK9	5.43427781230059E-07	0.108228822582565	0.502	0.293	0.0253742733889752	LDHA+	MAPK9
TIMMDC1	5.53692318633333E-07	0.129155847892121	0.621	0.387	0.0258535554339462	LDHA+	TIMMDC1
ANXA7	5.54298790901584E-07	0.265411785959087	0.931	0.796	0.0258818734435677	LDHA+	ANXA7

	p_val	avg_logFC	pct.1	pct.2	p_val_adj	Cluster	Gene
DDX39A	5.61808181956733E-07	0.183837413074553	0.781	0.529	0.0262325094401057	LDHA+	DDX39A
SNX17	5.83257971755076E-07	0.209743627491889	0.736	0.533	0.0272340644751597	LDHA+	SNX17
ALDOA	5.95772832454603E-07	0.224086845246093	0.99	0.907	0.0278184208658028	LDHA+	ALDOA
ERGIC3	6.07237247118011E-07	0.220203331789057	0.803	0.622	0.0283537287796813	LDHA+	ERGIC3
SRD5A3	6.63364836185074E-07	0.163684140892104	0.362	0.178	0.0309744942959896	LDHA+	SRD5A3
FUCA1	6.63701825935335E-07	0.198919286124487	0.468	0.262	0.0309902293583986	LDHA+	FUCA1
PMVK	6.72738559963604E-07	0.151783470170005	0.776	0.573	0.0314121815803805	LDHA+	PMVK
MVP	6.85372404595956E-07	0.231323442414841	0.672	0.467	0.032002093687799	LDHA+	MVP
PDHA1	6.85917705858688E-07	0.16489609863632	0.66	0.422	0.0320275554396597	LDHA+	PDHA1
GTPBP4	6.87460786618299E-07	0.196454449297198	0.685	0.476	0.0320996065095682	LDHA+	GTPBP4
TSG101	7.0457230917961E-07	0.177060722742795	0.835	0.649	0.0328985948325235	LDHA+	TSG101
POLD2	7.13414876255638E-07	0.182830721001611	0.687	0.449	0.0333114808170045	LDHA+	POLD2
XPNPEP1	7.40050415343631E-07	0.146693086472103	0.365	0.169	0.0345551740436402	LDHA+	XPNPEP1
TMEM208	7.46966294526018E-07	0.241038132416583	0.84	0.671	0.0348780971903034	LDHA+	TMEM208
GOLIM4	7.53638002468926E-07	0.126587990970842	0.756	0.564	0.0351896192492816	LDHA+	GOLIM4
TMEM259	7.57984090331223E-07	0.145946815380181	0.771	0.578	0.0353925511298358	LDHA+	TMEM259
SMCO4	7.6468019258824E-07	0.11624126631097	0.559	0.324	0.0357052122325227	LDHA+	SMCO4
GHITM	7.67835971791313E-07	0.242352968050424	0.948	0.769	0.0358525650308518	LDHA+	GHITM
YKT6	8.09437727297534E-07	0.116790891467512	0.67	0.467	0.0377950758007037	LDHA+	YKT6
NME1	8.47778334731831E-07	0.198480439103401	0.786	0.591	0.0395853137836334	LDHA+	NME1
SRI	8.70876261645168E-07	0.185508091477716	0.69	0.502	0.0406638252849978	LDHA+	SRI
NDUFB6	8.75474433893143E-07	0.167977852751141	0.909	0.76	0.0408785277417725	LDHA+	NDUFB6
PHB	8.83729587912794E-07	0.185332561493719	0.722	0.533	0.0412639856484121	LDHA+	PHB
TPRKB	9.07762348653612E-07	0.136590405556428	0.532	0.324	0.0423861473456831	LDHA+	TPRKB
IRAK1	9.28945216885029E-07	0.155090706491214	0.741	0.484	0.0433752390120127	LDHA+	IRAK1

	p_val	avg_logFC	pct.1	pct.2	p_val_adj	Cluster	Gene
SLC25A39	9.45007222769992E-07	0.225614910826517	0.8	0.631	0.0441252222527992	LDHA+	SLC25A39
B9D1	9.45558804876833E-07	0.149786976552228	0.596	0.378	0.044150977276114	LDHA+	B9D1
NTPCR	9.67986062907647E-07	0.127322626440974	0.421	0.222	0.0451981732353468	LDHA+	NTPCR
GNPAT	1.0313004053608E-06	0.134988200229868	0.564	0.333	0.048154509827512	LDHA+	GNPAT
TMEM128	1.05482556607762E-06	0.169073391489139	0.512	0.32	0.0492529701568624	LDHA+	TMEM128
ELOVL5	1.06716151601686E-06	0.185819173248671	0.704	0.542	0.0498289726673751	LDHA+	ELOVL5
ATG5	1.06784091679762E-06	0.236886195570028	0.495	0.302	0.0498606959280314	LDHA+	ATG5
MTRNR2L12	6.61394943608921E-07	0.203922625415595	0.991	0.99	0.0308825141019314	LDHA-	MTRNR2L12

Please see below the differentially expressed genes in LDHA-positive versus LDHA-negative α cells in T2D.

Supplementary Table 9b - LDHA-positive_vs_LDHA-negative α cells in T2D

	p_val	avg_logFC	pct.1	pct.2	p_val_adj	Cluster	Gene
LDHA	4.45643319195701E-34	1.76023619977461	1	0	2.08084235032049E-29	LDHA+	LDHA
LL22NC03-2H8.5	2.38959278583788E-08	0.195429912922578	0.541	0.132	0.00111577255949128	LDHA+	LL22NC03-2H8.5
FKBP1A	6.35783784466793E-07	0.349760233571848	0.861	0.566	0.0296866522481079	LDHA+	FKBP1A
FABP5	9.24287705143905E-07	0.333710481795161	0.762	0.382	0.0431577658162843	LDHA+	FABP5

We performed enrichment analysis to understand the major pathways that were overrepresented in the **LDHA-positive versus the LDHA-negative α cells**. We found that, almost uniformly, the overrepresentation referred to enrichment of genes related to metabolism and aerobic glycolysis (please see below top pathway enrichment according to curated libraries in Enrichr).

The Enrichr analysis is presented in the Supplementary Table 10 (see below).

Supplementary Table 10

NCI-Nature 2016 ⓘ p75(NTR)-mediated signaling Homo sapiens Validated targets of C-MYC transcriptional at ErbB1 downstream signaling Homo sapiens Signaling events mediated by PRL Homo sap HIF-1-alpha transcription factor network Ho	Panther 2016 ⓘ Glycolysis Homo sapiens P00024 ATP synthesis Homo sapiens P02721 De novo pyrimidine ribonucleotides biosyth De novo pyrimidine deoxyribonucleotide bic Huntington disease Homo sapiens P00029	BioPlex 2017 ⓘ RABGGTA ZFYE27 DNAI2 DCAF11 WDR77
huMAP ⓘ SLC25A5 NGRN PPP1R7 SRI MRPS30	PPI Hub Proteins ⓘ SLC2A4 GABARAPL2 SNCA GABARAPL1 MAP1LC3B	KEA 2015 ⓘ AAK1 PDK4 VRK2 PDK3 IRAK3
L1000 Kinase and GPCR Perturbations down ⓘ GABBR2 knockdown 96h HEPG2 ABL2 knockdown 96h HEPG2 GPR153 knockdown 96h HEPG2 PTK2B knockdown 96h HEPG2 GPRC5B knockdown 96h HEPG2	L1000 Kinase and GPCR Perturbations up ⓘ GPR83 knockdown 96h PC3 GRIN3A knockdown 96h HA1E GPR141 knockdown 96h PC3 IRAK2 knockdown 96h PC3 KISS1R knockdown 96h PC3	Kinase Perturbations from GEO down ⓘ TGFB2 knockout 296 GSE22989 AKT1 knockout 214 GSE39699 AKT1 activemutant 216 GSE9484 SYK druginhibition 290 GSE43510 MET knockout 252 GSE30651

Description No description available (169 genes)

BioPlanet 2019 ⓘ Tricarboxylic acid (TCA) cycle and respirato Mitochondrial protein import Electron transport chain Respiratory electron transport, ATP biosynth Protein metabolism	WikiPathway 2021 Human ⓘ Computational Model of Aerobic Glycolysis Cori Cycle WP1946 HIF1A and PPARG regulation of glycolysis WI Glycolysis and Gluconeogenesis WP534 Pyrimidine metabolism WP4022	KEGG 2021 Human ⓘ Diabetic cardiomyopathy Huntington disease Parkinson disease Prion disease Amyotrophic lateral sclerosis
ARCHS4 Kinases Coexp ⓘ CDK5 human kinase ARCHS4 coexpression CDK4 human kinase ARCHS4 coexpression BCKDK human kinase ARCHS4 coexpression PKMYT1 human kinase ARCHS4 coexpressio RPS6KB2 human kinase ARCHS4 coexpress	Elsevier Pathway Collection ⓘ Glycolysis Activation in Cancer (Warburg Effr Metabolic Reprogramming in Cancer: Over Glycolysis Omega-3-Fatty Acid Metabolism Insulin Resistance in Myocytes Induced by C	MSigDB Hallmark 2020 ⓘ Myc Targets V1 Oxidative Phosphorylation mTORC1 Signaling Protein Secretion Glycolysis
BioCarta 2016 ⓘ Endocytotic role of NDK, Phosphins and Dyr Downregulated of MTA-3 in ER-negative Bre: TSP-1 Induced Apoptosis in Microvascular Ei How does salmonella hijack a cell Homo sap Protein Kinase A at the Centrosome Homo s	Reactome 2016 ⓘ Metabolism Homo sapiens R-HSA-1430728 Metabolism of proteins Homo sapiens R-HS The citric acid (TCA) cycle and respiratory el Mitochondrial protein import Homo sapiens Respiratory electron transport, ATP synthesi	HumanCyc 2016 ⓘ pyrimidine deoxyribonucleotides biosynthes pyrimidine deoxyribonucleotides de novo bi superpathway of purine nucleotide salvage I purine nucleotides de novo biosynthesis Ho superpathway of conversion of glucose to a

Reviewer #2:

Thank you very much for the feedback that led us to design additional experiments and reconsider/strengthen our statements. In particular, your comments around the lingering HIF1 α expression and the possibility to be related to the IAPP/HFD background were reflected in the results from the new experiments and become a central point in the paper.

Q1. In figure 1, HIF1a levels remained increased in PFKFB3 beta KO. The authors mention that “these results indicated that PFKFB3 KO triggered a compensatory HIF1a expression in response to stress”. This statement is, to the reviewer’s opinion, not demonstrated by the data. Alternatively, it can also be independent of PFKFB3, and/or only related to the IAPP/HFD background.

A1. First, we hypothesized that HIF1 α expression was related to decompensation in response to the levels of DS. We have now repeated the experiment in mice at 7-10 weeks of age that had been exposed to eight weeks of a high-fat diet and introduced the fourth group of PFKFB3 β KO DS mice that had been on a chow diet for the final four weeks. Under Fig. 3a (below) is shown the experimental design.

Clearly, at lower levels of DS, the PFKFB3 β KO DS mice performed better metabolically than PFKFB3 β KO DS that were exposed to higher DS and their performance was comparable with that of WT mice in terms of the intra-peritoneal glucose tolerance test (IP-GTT). The fasting blood glucose levels of the PFKFB3 β KO DS mice were lower consistently than those of the PFKFB3 WT DS throughout the experiment from day 21 with significance at days 21, 31, 38 and 52.

When a chow diet was substituted for the high-fat diet in the PFKFB3 β KO DS mice, they performed even better as measured by the IP-GTT than did the WT controls on a high-fat diet. Interestingly, these results correlated with the expression of HIF1 α , which was significantly reduced in both PFKFB3 β KO DS mice and even more in PFKFB3 β KO DS mice chow diet compared with PFKFB3 WT DS mice.

These results indicate that the decompensation of the metabolic improvement after PFKFB3 knockout depends on the levels of HIF1 α expression. Increased HIF1 α immunolabelling is found to correlate with poor metabolic performance and vice versa, which indicates that PFKFB3 knockout can restore the function in regenerating β -cells only in the absence or at low expression levels of HIF1 α . **In marked manuscript text please refer to line 299-341 highlighted in grey and Figure Legend for Figure 3 and 4.**

Please see the figure below (Figure 4), which shows HIF1 α expression under lower DS (age 7-10 weeks and eight weeks of high-fat diet).

In relation to the origin of HIF1 α expression, we addressed the type of damage that marks lower (HIF1 α expression is diminished) or higher diabetogenic stress (HIF1 α expression is high).

We found that cytoplasmic c-Myc (calpain activation) was not detectable at lower DS [Supplementary Figure 15a, positive control PFKFB3 WT DS (13 weeks) at far right] and instead that ER stress marker CHOP was elevated in the PFKFB3 WT DS group but was abated in

PFKFB3 β KO DS. These results indicated that HIF1 α expression at higher DS correlated with misfolded protein stress (calpain activation). In marked manuscript text please refer to line 511-518 highlighted in grey and Figure Legend for Supplementary Figure 15b,c. Please see the CHOP immunostaining presented below in b and c.

Q2. Concerning the metabolic phenotype of these mice, it is not clear why the authors presented fasting insulin/beta-cell mass. This is not commonly presented. The insulin levels during IPGTT are missing. This is probably more informative and directly related to in vivo beta cell function.

A2. We have now corrected the insulin presentation as suggested. We have introduced IP-GTT insulin values in the repeated experiment after eight weeks of exposure to a high-fat diet (lower levels of DS). In the PFKFB3 WT DS mice, levels of plasma insulin dropped 30 minutes after glucose stimulation, the levels either did not change or were slightly increased in the other groups, including the PFKFB3 β KO DS mice. In marked manuscript text please refer to line 323-326 highlighted in grey and Figure 3h and below.

Q3. The number of mice used is limited also, and should be increased to get more statistical power.

A3. We performed the experiment after eight weeks exposure to a high-fat diet, which is now introduced in the manuscript and explained in A2 to Reviewer #2. In marked manuscript text please refer to line 299-341 highlighted in grey and Figure Legend for Figure 3.

Q4a. The single cell analysis is confusing. The authors did not mention where to which cluster PFKFB3 RNA belongs, if it is detected.

A4a. We focused on β -cells that expressed HIF1 α , since this β -cell subpopulation seemed to predict the restoration of functional competence of β -cells in PFKFB3 β KO DS mice.

PFKFB3 levels were low to undetectable and therefore they could not be used for interpretation. Please see below the measured expression levels of PFKFB3 to compare LDHA-positive and LDHA-negative β -cells (left) in healthy non-diabetics (ND) and the same cells from donors with T2D (right) (Supplementary Fig. 9 and below).

We believe that these low expression levels reflect the post-translational stabilisation of PFKFB3 in response to hyperglycaemia post-onset of T2D. This is unlike prediabetes that we previously tested (Montemurro et al., 2019) and may explain the low levels of detectable gene expression in human pancreas samples. In support of this, PFKFB3 and HIF1 α were found to be significantly upregulated in T2D by proteomics and not in transcriptomics in the newest *Nature Metabolism* paper by Mathias Man and colleagues (<https://pubmed.ncbi.nlm.nih.gov/34183850/>).

Q4b. Then the conclusion related to the data mining of scRNA seq are interesting, but it is difficult to directly link to PFKFB3 KO mice results. We know that loss of identity appears during T2D, but whether this is related to PFKFB3 remains unknown, and at least, scRNA seq does not prove anything here. The reviewer does not see the link between scRNA seq data presented here, and the scope of the paper, that is to demonstrate that PFKFB3 beta cell specific deletion increases beta cell proliferation. The fact that bihormonal cells exist in T2D human beta cells and PFKFB3

pancreas does not mean that it is directly related. This should be proven by doing scRNA seq in WT and KO mice, but this is probably beyond the scope of this paper. I thus suggest to write the scRNA seq part to make it more comprehensible for the reader.

A4b. The scRNA Seq part was rewritten and explained using the LDHA surrogate marker for the HIF1 α -fated β -cell population to explain how the HIF1 α -positive subpopulation contributed to β -cell dysfunction. This was done so that the persistent expression of HIF1 α after PFKFB3 knockout under high levels of DS could be linked to glucose intolerance. At the same time, human β -cells with an HIF1 α signature were bihormonal; since they expressed LDHA, which is a disallowed gene, this β -cell subpopulation was deemed dysfunctional. In marked manuscript text please refer to line 352-389 highlighted in grey.

Q5. In the material & methods section, it is mentioned that « Retro-orbital bleeding was used to collect the blood for the second IP-GTT prior to and 30 minutes after glucose bolus injection. The mice were anaesthetized by brief exposure to isoflurane (10 seconds). » This is not common to anaesthetize mice during an IP-GTT, since isoflurane may affect insulinemia and glycemia could happen. Can you comment on that?

A5. We understand the reviewer's concern regarding use of isoflurane. We followed the protocol for a very brief induction of 1% isoflurane (10 sec). As described in <https://www.ncbi.nlm.nih.gov/pmc/articles/PMC4908499/pdf/PHY2-4-e12824.pdf>, the presence of isoflurane significantly increases blood-glucose levels without affecting insulin when administered continuously, first at 1% for induction and then at 3% for maintenance. Also, in the experiment with mice under lower DS (8 weeks exposure to HFD at younger age), we performed carotid artery bleeding without use of isoflurane. The results for insulin at 0 min and 30 min in terms of IP-GTT are presented here (Figure 3h, see below). In marked manuscript text please refer to line 138-144 highlighted in grey.

Q6. In addition, it is mentioned that « When blood glucose exceeded the detection range of the blood glucose meter, plasma glucose was determined using the glucose oxidase method and analyzed with YSI 2300 STAT PLUS Glucose and L-Lactate Analyzer ». It may be difficult to

interpret the results when two different methods are used to measure glycemia. Can you also comment on that?

A6. To relate the values given by the glucometer and the YSI 2300 STAT PLUS Glucose and L-Lactate analyser, we contacted the manufacturers and relied on their notes and protocols when we validated the analyser relative to the glucometer. We only used YSI 2300 STAT PLUS Glucose and L-Lactate analyser for the plasma samples from the IP-GTT at 12 weeks after HFD in the experimental paradigm for high diabetogenic stress (DS).

We also selected randomly 13 mice, for which we determined whole-blood and plasma-glucose levels, and we found the average ratio for correction to convert the values from plasma glucose to whole-blood glucose (0.56). Below is the table with the values from which we derived the conversion factor 0.56 (plasma to whole blood).

Supplementary Table 1 – conversion coefficient derivation from plasma- to whole blood glucose

Mouse ID	Body weight (g)	Whole blood glucose (mg/dl)	Plasma glucose (mg/dl)	Plasma/Whole blood glucose ratio
1618	23.9	162	333	2.055555556
1617	25.2	134	207	1.544776119
1616	25	95	212	2.231578947
1615	23.9	142	171	1.204225352
1608	29.4	115	199	1.730434783
1609	29.3	140	331	2.364285714
1610	27.5	109	227	2.082568807
1612	28.5	128	292	2.28125
1642	26.3	153	200	1.307189542
1643	28.8	102	205	2.009803922
1644	24.5	153	166	1.08496732
1657	22.6	156	261	1.673076923
1658	24.8	112	178	1.589285714
			DIFFERENCE	1.781461438

Minor points

Q1. The genetic background of the mice used in this study should be mentioned, as well as the tamoxifen injection protocol (dose, number of IP injections) and the reference of the HFD.

A1. We have edited the paper to explain the genetic background of the mice (FVB) and we have detailed the tamoxifen injections and the reference for the high-fat diet (PMCID: PMC4792296). In marked manuscript text please refer to line 99 and 109-113 highlighted in grey, respectively.

Reviewer #3 (Remarks to the Author):

We thank you for your attention and precision in reading and assessing our manuscript. That has enormously improved the reading and the quality of the manuscript, for which we are very grateful. We hope to have answered your questions and comments to satisfaction of your justified critique.

Q1. The title in the manuscript needs to be correct.

A1. We have changed the title of the manuscript to:

“ β -cell-specific deletion of *PFKFB3* restores β -cell fitness competition yielding healthy β -cell replication under diabetogenic stress”

Q2. Please, provide the amount of glucose bolus injected during the IP-GTT.

A2. Now we have provided the amount of glucose bolus in our manuscript; it is 2g/kg of body weight. In marked manuscript text please refer to line 136-138 highlighted in grey.

Q3. In the methods section (“Insulin and glucose tolerance tests”), the authors only describe the glucose tolerance test (IP-GTT). Insulin tolerance test (IP-ITT) is described in “Glucose and insulin assays”.

A3. Now we have provided a full description of the IP-GTT and IP-ITT. Please see below:

“An intraperitoneal glucose tolerance test (IP-GTT) was performed at 9 and 12 weeks after the start of the HFD (19 and 22 weeks after the tamoxifen injection) for high DS mice and at 4-, 5- and 6 weeks after start of the HFD for lower DS mice. Mice were fasted overnight in a clean cage and with access to water before the tests. Tail vein blood glucose was collected before and 15, 30, 60, 90 and 120 minutes after a 20% glucose bolus injection (2g/kg of body weight). Retro-orbital or carotid bleeding was used to collect the blood prior to and 30 minutes after the glucose bolus injection, at 12 weeks and 8 weeks of HFD for high and low diabetogenic mice respectively. For retro-orbital bleeding, mice were anaesthetised by brief exposure to isoflurane (10 seconds). Carotid bleeding was performed on the non-anesthetized mice in lower DS group. The blood was collected in a microcentrifuge tube coated in ethylenediamine tetra acetic acid (EDTA) buffer and the plasma was obtained by centrifuging the samples for 10 minutes (5000 RCF, 10min, 4°C). An intraperitoneal insulin tolerance test (IP-ITT) was performed in conscious mice 51 and 56 days after beginning of HFD, for high and low diabetogenic stress, respectively. The mice were fasted for six hours prior to the test. To measure glucose levels, tail vein blood was collected prior to and 20, 40 and 60 minutes after administration of insulin (0.75 IU/kg of body weight) (Lilly insulin Lispro, LLC, Indianapolis, USA).”

In marked manuscript text please refer to line 132-149 highlighted in grey.

Q4. Why IP-GTT was assess after 19 and 22 week and IP-ITT was assess only after 19 weeks from tamoxifen injection, as described in the methods? Why the authors did not show the results from 22 weeks?

A4. We did not perform the ITT at 22 weeks (12 weeks after HFD).

Please see below the results from IP-GTT performed at 22 weeks and now presented in Supplementary Figure 3 (see below).

We performed the ITT in the experiment that was conducted at lower levels of DS (8 weeks exposure to high-fat diet in younger mice) and the results are presented below and in Figure 3f.

Q5. “Data are presented as an error of the mean (standard error, SEM) for the number of mice indicated.” Please, added “indicated in the figure’s legends”.

A5. We have now added “indicated in the figure’s legends” – please see the marked manuscript text in line 240-241 in grey

Q6. The “statistical analyses” section should be better described. The authors could explain with more detail de calculation of AUC using the trapezoidal rule or cite a reference that explain it. Also, it was described that Student’s t-test was used to compare the mean data between groups, but all experiments presented in the manuscript have three groups. ANOVA one-way followed by a post-hoc test should be used to compare three groups.

A6. Now we have altered the paper to describe in detail the AUC calculation and have cited a reference that explains this calculation using the trapezoidal rule (Allison DB, Paultre F, Maggio C, Mezzitis N, Pi-Sunyer FX. The use of areas under curves in diabetes research. *Diabetes Care*. 1995 Feb; 18(2):245-50. doi: 10.2337/diacare.18.2.245. PMID: 7729306). We used Prism software (version 8.2.1) and followed the instructions to compute the AUC by the trapezoidal

method (<https://www.graphpad.com/support/faq/how-does-prism-compute-area-under-the-curve-auc-what-are-its-units/>).

We also applied an one-way analysis of variance (one-way ANOVA) followed by Tukey post-hoc test. Please see the marked manuscript text in line 239-245 highlighted in grey.

Q7. Please, provide the fasting insulin and c-peptide levels as you provided fasting glucagon levels (Figure 2F).

A7. Now we have provided the levels of fasting insulin and c-peptide levels after IP-GTT, please see below and Figure 2g,e.

Q8a. Is it correct using “fasting insulin/beta-cell mass” to refer to “fasting insulin/c-peptide”? The authors should know that lower insulin/c-peptide ratio may suggest an increase in the insulin clearance (Piccinini and Bergman. The Measurement of Insulin Clearance. Diabetes Care 2020 Sep; 43(9): 2296-2302.). Thus, PFKFB3 beta KO DS mice seem to display an increase in insulin clearance process, which could impact plasma insulin levels.

A8a. This is a great point, and we thank the reviewer. We have now corrected the presentation of the insulin data. We have reflected on the insulin clearance, as kindly guided by the reviewer, and we have used this ratio to indicate insulin clearance during IP-GTT testing (Figure 2h). No significant differences were detected among the groups.

Q9. Based on the results from Figure 2, the authors suggest impaired insulin secretion in the PFKFB beta KO DS mice. However, the results displayed in this figure did not support this suggestion.

The authors should, at least, show the c-peptide levels during the IP-GTT, or evaluating insulin secretion with more directly methods (for example, glucose stimulated insulin secretion in isolated islets).

A9. The plasma insulin after IP-GTT in mice at lower diabetogenic stress (DS) is presented in Fig. 3h (please see below).

The c-peptide and insulin levels from experiment under high diabetogenic stress (13 weeks) are presented now (Figure 2g,e)

REVIEWERS' COMMENTS:

Reviewer #1 (Remarks to the Author):

In the revised manuscript, the authors included additional experimental data and single-cell RNA-seq analysis. The authors addressed most of my questions. My lingering question is in the single-cell RNA-seq analysis. The authors separated the human beta cell groups based on their HIF1 α activity, but does these two groups correlated with differential PFKFB3 expression? The claim of "Insulin and glucagon double positive cells in diabetic mice resemble human beta cells with HIF1 α signature" seems to be unfounded since there is no confirmation of the signatures in the mouse bihormonal cells.

Reviewer #2 (Remarks to the Author):

The authors have included new sets of data, replicated key experiments and replied to reviewer's concerns. The manuscript has been improved now and demonstrate the key role of PFKFB3 in beta cell function during metabolic stress.

The recovery experiment is really important and suggests that the level of HIF1 α in beta cells is key to restore beta cell function.

POINT-BY-POINT ANSWERS TO REVIEWER 1 COMMENTS AND CONCERNS

Thank you for reviewing our manuscript entitled " **β -cell-specific deletion of PFKFB3 restores cell fitness competition and physiological replication under diabetogenic stress**" - COMMSBIO-21-1041B

Here are our point-by-point answers to reviewers' comments:

Reviewer #1

Thank you very much for the query for the additional clarifications.

Q1. My lingering question is in the single-cell RNA-seq analysis. The authors separated the human beta cell groups based on their HIF1 α activity, but does these two groups correlated with differential PFKFB3 expression?

A1. We focused on the β -cells that expressed HIF1 α , since the presence of this β -cell subpopulation seemed to predict the restoration of functional competence of β -cells in PFKFB3^{βKO} DS mice. In our experimental design reproducing the low and high diabetogenic stress (high and low DS), HIF1 α expression in PFKFB3^{βKO} DS mice was independent of PFKFB3 at high DS.

PFKFB3 expression levels data were low to undetectable according to the sc RNA-Seq analysis and therefore they could not be used for correlating the HIF1 α (LDHA) positive- versus HIF1 α (LDHA) negative cells. However, phosphofructokinase 1 platelet type (PFKFP), the PFKFB3 target of allosteric activation attuning aerobic glycolysis to increased flux co-clustered with LDHA (HIF1 α)-positive cells and this extended to couple of other markers of aerobic glycolysis such as glucokinase (GK), and pyruvate dehydrogenase kinase 4 (PDK4), hexokinases

1 and 3 (HK1 and 3) and SLC2A1 (GLUT1) indicating that LDHA (HIF1 α)-positive cells were metabolically remodeled.

The expression levels of PFKFB3 in LDHA (HIF1 α) -positive and LDHA (HIF1 α)-negative β -cells (left) in healthy non-diabetics (ND) and in donors with T2D (right) was included in the Supplementary Fig. 9 a,b and above.

The low gene expression levels of PFKFB3 may reflect the post-translational stabilisation of PFKFB3 in response to hyperglycaemia post-onset of T2D in humans or in the early onset of T2D in our current mouse model, decoupling PFKFB3 levels from the HIF1 α transcriptional activation. This is unlike in prediabetes that we previously tested (Montemurro et al., 2019) where we reported that PFKFB3 was transcriptionally upregulated in a HIF1 α -dependent fashion. The low levels of detectable PFKFB3 gene expression in human pancreas samples were also indicated in the recent *Nature Metabolism* paper by Mathias Man and colleagues (<https://pubmed.ncbi.nlm.nih.gov/34183850/>) where both PFKFB3 and HIF1 α were found to be significantly upregulated in T2D on protein level by proteomics and were not selected from the transcriptomics analysis. Now we introduced a comment on PFKFB3 expression levels in the results and the discussion (please see lines 225-227 and 400-402 marked in grey).

Q1. The claim of “Insulin and glucagon double positive cells in diabetic mice resemble human beta cells with HIF1a signature” seems to be unfounded since there is no confirmation of the signatures in the mouse bihormonal cells.

A1. We agree with the reviewer and now we rephrased and clarified our statements in the Discussion section. We now acknowledge that the bihormonal cells that were observed in PFKFB3^{WT} DS mice share the same bihormonal status- but may differ in the expression of disallowed genes that confers a dysfunctional phenotype of the mouse HIF1 α -positive β -cells and human LDHA (HIF1 α)-positive β -cells in T2D. Please see our rephrased statements in the lines 190-191 and 407-411 marked in grey.

“Importantly, mouse HIF1 α -positive β -cells differed from the human *LDHA*-positive cells with HIF1 α signature and bihormonal status, because they persisted in PFKFB3^{βKO} DS mice under high DS where bihormonal cells were depleted. This further highlighted the specific role that PFKFB3 and not HIF1 α plays in the generation or sustenance of the bihormonal cells.”

Reviewer #2:

Thank you very much for the constructive review that improved our paper and the encouraging feedback on the study.